# Nanoscale patterning of collagens in *C. elegans* apical extracellular matrix

Jennifer R. G. Adams [1,4], Murugesan Pooranachithra[1,4], Erin M. Jyo[1], Sherry Li Zheng [1], Alexandr Goncharov[2], Jennifer R. Crew[3], James M. Kramer[3], Yishi Jin[2], Andreas M. Ernst[1] & Andrew D. Chisholm [1,2] ✉

Apical extracellular matrices (aECMs) are complex extracellular compartments that form important interfaces between animals and their environment. In the adult *C. elegans* cuticle, layers are connected by regularly spaced columnar structures known as struts. Defects in struts result in swelling of the fluid-filled medial cuticle layer ('blistering', Bli). Here we show that three cuticle collagens BLI-1, BLI-2, and BLI-6, play key roles in struts. BLI-1 and BLI-2 are essential for strut formation whereas activating mutations in BLI-6 disrupt strut formation. BLI-1, BLI-2, and BLI-6 precisely colocalize to arrays of puncta in the adult cuticle, corresponding to struts, initially deposited in diffuse stripes adjacent to cuticle furrows. They eventually exhibit tube-like morphology, with the basal ends of BLI-containing struts contact regularly spaced holes in the cuticle. Genetic interaction studies indicate that BLI strut patterning involves interactions with other cuticle components. Our results reveal strut formation as a tractable example of precise aECM patterning at the nanoscale.

Apical extracellular matrix (aECM) is found throughout the animal kingdom and forms a variety of structures that play key roles in mediating interactions between organisms and their environment. Well-studied examples of aECM include the collagenous cuticle of nematodes, the chitinous cuticles of arthropods, and sensory structures such as the tectorial membrane of the vertebrate inner ear[1]. Compared to highly conserved basal ECM structures such as basement membranes[2–4], aECMs are more diverse in structure, function, and molecular composition. Many aECMs display complex three-dimensional architecture and are intricately patterned at the micro and nano scales[5,6]. In certain cases aECM patterning has been directly related to cytoskeletal pattern in the underlying epithelium[7,8]. However in general we understand very little about how complex patterns are generated in aECM.

The cuticle of the nematode *C. elegans* is a classic example of an aECM that has long been studied for its function in body and skin morphology[9,10]. The cuticle also plays a critical role in permeability barrier function[11], host-pathogen interactions[12], auditory sensation[13],

and sensation of stress or physical damage[14]. *C. elegans* generates distinct cuticles in each larval stage; the mature adult cuticle is generated in the final (L4) larval stage[15]. The cuticle is composed of collagens and a wide variety of non-collagenous proteins[16]. Cuticle collagens themselves form a large multigene family with approximately 178 members in *C. elegans*; the functions of about 15% of cuticle collagens have so far been defined by genetic analysis[17]. Molecular and transcriptomic studies indicate that collagen expression is precisely regulated during each larval stage and in adulthood[18–20], presumably related to the stage and region-specific 3D architecture of the aECM. Studies of specific collagens have revealed they localize to specific substructures within the cuticle, such as furrows and annuli[21]. How specific collagens are patterned within the aECM to generate its intricate architecture remains elusive.

The *C. elegans* adult cuticle is the most structurally complex cuticle type (https://www.wormatlas.org/hermaphrodite/cuticle/Cutframeset.html) and is generated in the mid to late L4 stage. The dorsolateral and ventrolateral adult cuticle, generated by the hyp7 syncytium, contains

[1]Department of Cell and Developmental Biology, School of Biological Sciences, University of California San Diego, La Jolla, CA 92093, USA. [2]Department of Neurobiology, School of Biological Sciences, University of California San Diego, La Jolla, CA 92093, USA. [3]Northwestern University School of Medicine, Department of Cell and Molecular Biology, Chicago, IL 60611, USA. [4]These authors contributed equally: Jennifer R. G. Adams, Murugesan Pooranachithra. ✉e-mail: adchisholm@ucsd.edu

circumferential ridges (annuli) separated by furrows. The lateral adult cuticle, generated by underlying seam cells, contains lateral longitudinal ridges known as alae[8]. Unlike larval cuticles, the adult cuticle also contains a fluid-filled medial layer between its outer (cortical) and inner (basal) layers, within which are ordered columnar structures known as struts[16,22]. Struts likely provide mechanically strong yet flexible attachments between cortical and basal layers of the adult cuticle. Loss of function in struts causes separation of cuticle layers and expansion of the fluid-filled medial layer, the Blistered (Bli) phenotype[23]. Genetic screens for cuticle morphology defects have yielded six *bli* genes divided into two molecular classes: three cuticle collagens encoded by *bli-1*, *bli-2*, and *bli-6* (this study), and three cuticle processing enzymes encoded by *bli-3*, *bli-4*, and *bli-5*[24–27].

Here we focus on the function, localization, and interactions of the cuticle collagens encoded by *bli-1*, *bli-2*, and *bli-6*. We define the null phenotypes of these BLI collagens and explore their interactions with each other and with other cuticle components. Using knock-ins to endogenous loci we show that all three BLI proteins are initially expressed in the L4 stage and become localized to cuticle struts in

the nascent adult cuticle. We examine the biogenesis of strut puncta and how their patterning depends on other cuticular collagens. Using 3D-structured illumination super-resolution microscopy (3D-SIM) we find evidence for nanoscale organization of struts, revealing previously unknown levels of patterning in the cuticle. Our findings provide insight into how complex pattern is generated in an apical extracellular matrix.

## Results

### Mutations in three BLI cuticle collagens cause adult-onset blistering

The blistering (Bli) phenotype denotes the appearance of fluid-filled swellings within the cuticle known as blisters (Fig. 1a). As detailed below, mutations in collagens *bli-1*, *bli-2*, and *bli-6* cause adult-specific Bli phenotypes ranging from mild to severe. Mild blistering is defined as small blisters on the nose and/or tail, or one medium-sized blister in the body; intermediate is defined as several medium-sized blisters, and severe blistering is defined as large blisters extending along the entire body, leading to defects in egg laying,

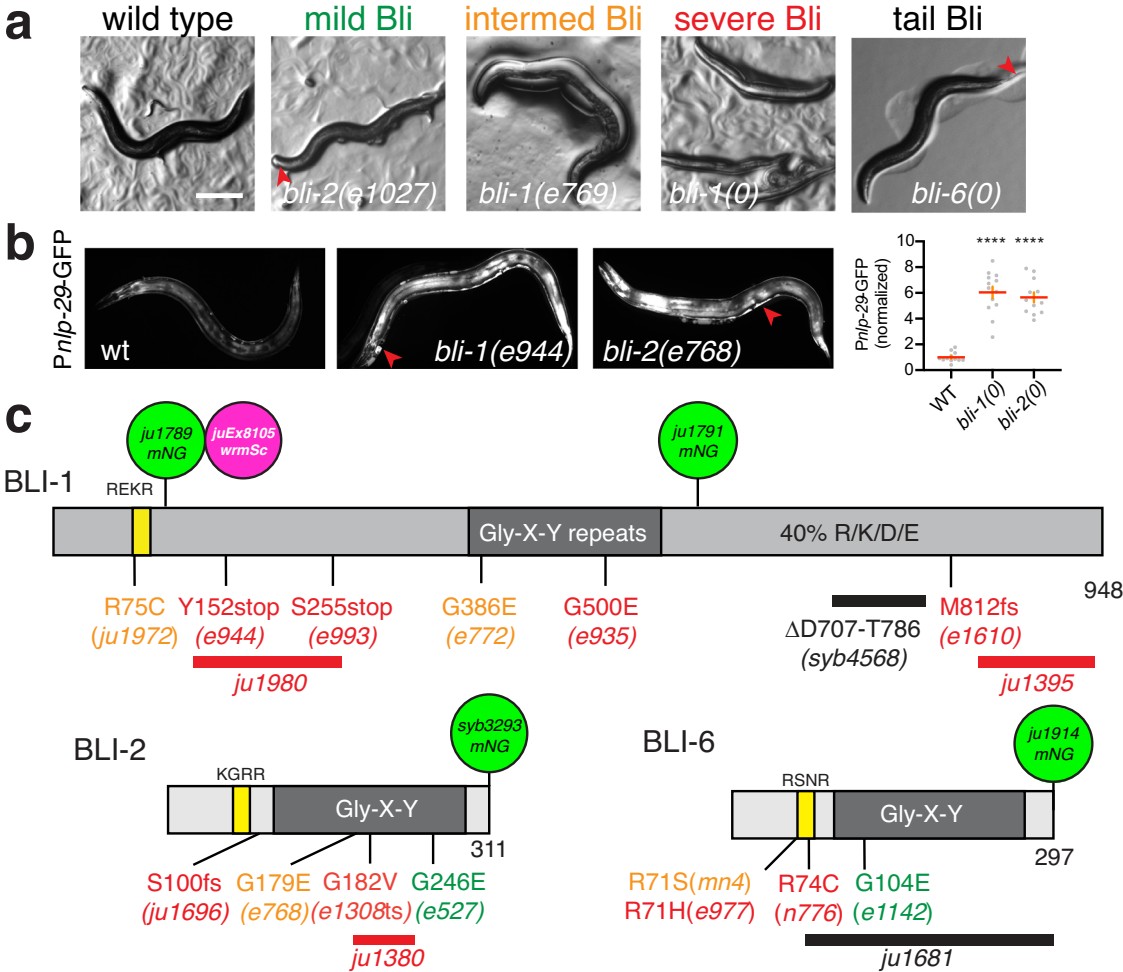

**Fig. 1 | Visible phenotypes of *bli* mutants and molecular genetics. a** The Bli (Blistered phenotype). Dissection microscope images of representative adult animals: wild type; mild Bli, intermediate Bli, and severe Bli; for definitions see the text. *bli-6(0)* animals are mostly non-Bli but display small blisters at the tail (red arrowhead). Scale, 100 μm. **b** Upregulation of epidermal innate immune response in Bli mutants, shown by P*nlp-29*-GFP(*frIs7*). Note GFP leakage into cuticle blisters (red arrowheads). Widefield images of adults within 12 h of L4 stage. Scale 100 μm. Quantitation of P*nlp-29*-GFP fluorescence, normalized arbitrary units, showing mean (red lines) and SEM (orange error bars). Statistics, ordinary one-way ANOVA,

Dunnett's post test; ****$P < 0.0001$. **c** Predicted protein structure of three BLI collagens, cartoons with N termini to the left. Predicted subtilisin-like cleavage sites (RXXR motif, also known as Homology Box A) are yellow boxes and the Gly-X-Y repeats shown in dark gray. The C-terminal region of BLI-1 is enriched in charged residues (~20% Arg or Lys, ~20% Asp or Glu) but has no other detectable sequence motifs. Mutations causing mild, intermediate, or severe phenotypes are indicated by the color code as in (**a**). Additional *bli* mutations are listed in Tables 1–3. Uniprot entries: BLI-1 (Q09457), BLI-2 (O01914), BLI-6 (Q8MXR1).

**Table 1 | Phenotypic and molecular characterization of *bli-1* alleles**

| Allele | Mutagen | Bli severity | DNA change | Predicted protein change |
|---|---|---|---|---|
| *e769*[a] | EMS | Intermediate /severe | Exon 4 splice donor | |
| *e771* | EMS | Non-Bli | AGA > AAA and exon 4 splice donor (CAGgta > CAGata) | R748K |
| *e772* | EMS | Intermediate /severe | GGG > GAG | G386E |
| *e935* | EMS | Severe | GGA > GAA | G500E |
| *e944* | [32]P | Severe | TAC > TAA | Y152stop |
| *e993* | spo | Severe | TCA > TAA | S255stop |
| *e1203* | EMS | Severe | 1 bp insertion | Frameshift at G343 |
| *e1431* | EMS | Intermediate | Exon 2 splice acceptor (tagGAT > taaGAT) | |
| *e1610* | spo | Severe | 4 bp deletion | Frameshift at M812 |
| *ju1395* | CRISPR | Severe | 645 bp deletion + 42 bp insertion | Deletion of last 110 aa (P839 onward) |
| *ju1874* | CRISPR | Severe | 2 bp deletion | R585 frameshift |
| *syb4568* | CRISPR | Non-Bli | 240 bp deletion | In-frame deletion of 80 aa (D707-T786) |
| *ju1972* | CRISPR | 22% severe Bli, 25% intermediate or mild | CGT > TGT | R75C |
| *ju1980* | CRISPR | Severe | 417 bp deletion | In-frame deletion of 139 aa (R125-S263) |

[a]Same lesion as found in *e976*. Alleles *m361*, *sc63*, and *st421* were not available for reanalysis; *m361* had been reported to cause a premature stop in the N-terminus. The *ju1395* deletion affected 113 bp in exon 5 of *dph-2*. Putative *bli-1* null alleles such as *ju1395* or *ju1980* displayed 100% penetrant severe Bli phenotypes by 24 h post L4 stage (*n* = 50 scored per genotype).

locomotion, feeding, and premature death (Fig. 1a; Tables 1–3). In the most severe *bli-1* or *bli-2* mutants small blisters are first visible within 3 h of the L4/adult molt and subsequently combine to form larger blisters, leading to severe blistering by ~9 h after the L4/adult molt, with 100% penetrance (*n* > 50). Severely blistered animals are fertile and typically lay a small number of eggs prior to death. Reduction in *bli-1* function via RNAi induces epidermal responses to injury such as upregulation of the antimicrobial peptide *nlp-29*;[28] we confirmed this *nlp-29* upregulation in young adults using *bli-1* and *bli-2* mutants (Fig. 1b). Thus loss of function in the BLI collagens results in adult-onset defects in cuticle integrity and consequent epidermal damage responses.

*bli-1*, *bli-2*, and *bli-6* encode nematode cuticle collagens, defined by their central Gly-X-Y domains and characteristic N-terminal sequence motifs (Fig. 1c). BLI-2 and BLI-6 collagens are typical nematode cuticle collagens, whereas BLI-1 is unusually large with extended N- and C-terminal domains that are conserved in other nematodes of the *Caenorhabditis* genus (Fig. S1). Three *bli-1* mutations (*e944*, *e993*, and *m361*) result in premature stop codons and cause recessive severe Bli phenotypes. Four *bli-1* mutations affect splice sites (*e769*, *e771*, *e976*, and *e1431*) and three are predicted to cause frameshifts (*e1203*, *e1610*, and the CRISPR/Cas9-engineered deletion *ju1395*). Several *bli-1* mutations cause missense alterations in the Gly-X-Y repeats, including the severe recessive Bli allele *e935*; Gly substitutions in the Gly-X-Y repeats have been frequently found in *C. elegans* cuticle collagen mutants, and are thought to interfere with the assembly of the nascent collagen, resulting in loss or gain of function[29,30]. We conclude the *bli-1* null phenotype is a recessive severe adult Bli and below refer to *ju1395* as *bli-1(0)*. To test the role of the extended N-terminal and C-terminal domains in BLI-1 we engineered in-frame deletions or point mutations using CRISPR/Cas9. An in-frame deletion of 80 residues in the charged C-terminus (*syb4568*) did not cause overt Bli phenotypes (Table 1), whereas an in-frame deletion within the N-terminus (*ju1980*) displayed a severe Bli phenotype, suggesting this region of the N-terminus is important for BLI-1 function.

Several *bli-2* alleles cause substitutions in the Gly-X-Y repeats and result in weak or intermediate Bli phenotypes respectively (Table 2). We used CRISPR/Cas9 to engineer an in-frame deletion *bli-2(ju1380)* in the Gly-X-Y repeats and found that this caused a severe Bli phenotype, consistent with this being the *bli-2* null phenotype; *ju1380* is referred to below as *bli-2(0)*. All the above alleles of *bli-1* and *bli-2* were recessive to wild type.

Multiple *bli-6* alleles cause dominant or semidominant severe Bli phenotypes and affect Arg residues in the putative subtilisin cleavage sequence (RSNR, residues 71–74 in BLI-6; Fig. 1c). The alleles *sc16* and *e977* cause R71H, the semidominant alleles *n776* and *e519* cause R74C, and the fully dominant allele *mn4* causes R74S (Table 3). Similar missense alterations of subtilisin cleavage sites in other cuticle collagens frequently cause dominant gain of function[29]. We therefore engineered a similar change in BLI-1 (R75C, *ju1972*) and found that this caused a dominant intermediate Bli phenotype, consistent with BLI-1 also being cleaved at the RXXR site. Animals homozygous for the deletion *bli-6(ju1681)* developed small blisters in the tail region (Fig. 1a) but were otherwise normal in morphology. Thus *bli-6* may normally function redundantly in cuticle morphology; altered BLI-6 processing in gain of function mutants may interfere with function of other collagens.

### *bli* collagen mutants display specific defects in strut formation

To better understand the origin of the cuticle defects in *bli* mutants we analyzed cuticle ultrastructure by transmission electron microscopy (TEM). The adult cuticle consists of three major layers: an external cortical layer, a medial layer, and a basal layer composed of multiple sublayers including two fibrous layers (Fig. 2a). In contrast to *C. elegans* larval cuticles, the adult cuticle has a mostly electron-lucent medial layer containing electron-dense pillar-like structures known as struts[15,31]. Struts are pillar-like structures oriented perpendicular to the plane of the cuticle and appear to connect the cortical and basal layers. In our EM data of wild-type animals the medial layer is 221 ± 11.7 nm thick in young adults (Fig. 2e); struts are 193 ± 7 nm in width and spaced 518 ± 32 nm apart (*n* = 35) along circumferential axes.

All *bli* mutants examined displayed aberrant or missing struts in TEM; for example no struts were observed in >500 sections of *bli-1(e944)* or *bli-2(e768)* mutants and in >300 sections of *bli-6(n776)* (Fig. 2b–d). Cortical and basal cuticle layers were typically recognizable but sometimes aberrant, suggesting the Bli phenotype reflects an initially specific disruption of strut or medial layer formation that may lead to secondary disruption of other layers. In blistered regions the cortical layer was thinner than in wild type (Fig. 2e). In *bli-1(lf)*, *bli-2(lf)*, and *bli-6(gf)* mutants the medial layer was variably expanded in thickness; such blistered areas lacked struts and instead contained loose electron-dense granular or occasionally fibrous material (Fig. 2b–d), consistent with a study of *bli-1(m361)*[32]. The cortical and basal layers were typically separated in these mutants, even in areas without overt

**Table 2 | Phenotypic and molecular characterization of *bli-2* alleles**

| Allele | Mutagen | Bli severity | DNA change | Predicted protein change |
|---|---|---|---|---|
| *e107* | EMS | Severe | GGA > AGA | G127R |
| *e527*ts | EMS | Variable (ts) | GGA > GAA | G246E |
| *e768* | EMS | Intermediate/severe | GGA > GAA | G179E |
| *e1027* | EMS | Mild | CAGgtaatt g > a in intron 1 | |
| *e1308*ts | EMS | Non-Bli at 20 °C | GGA > GTA | G182V |
| *ju1380* | CRISPR | Severe | 120 bp deletion + 3 bp insertion | In-frame deletion of L198-Q236 |
| *ju1696* | CRISPR | Severe | 10 bp insertion | S100 > FE*STOP |
| *ju1697* | CRISPR | Non-Bli | 21 bp insertion | S100 > FKCQMPNA |

*e527* animals were mostly non-Bli at 15 °C. *e1308* mutants were rarely Bli at 15 °C or 20 °C and severely Bli at 25 °C. The *st1016* allele was not available for reanalysis, but also caused a G > R missense alteration in the Gly-X-Y domain. $n = 50$ scored per genotype; severe mutants were fully penetrant Bli.

blisters, although in some areas the cortical and basal layers were adjacent. Other than these defects, the other layers of the cuticle appeared mostly normal; in *bli-6(gf)* mutants, abnormal electron-dense vesicle-like structures of unknown origin were seen in the basal layer (Fig. 2d).

### Endogenously tagged BLI collagens localize to puncta corresponding to cuticle struts

We next examined the localization of the BLI collagens using fluorescent protein knock-ins to endogenous loci, inserted either N-terminal or C-terminal to the Gly-X-Y repeats or at the C-terminus (Fig. 1c; see Methods). BLI-1::mNG fusion proteins were expressed exclusively in the late L4 epidermis and in the nascent and mature adult cuticle (Fig. 3a). Within the cuticle, BLI-1::mNG was localized to periodically repeating puncta of 0.25–0.3 μm in diameter in circumferential rows that extended from the dorsal or ventral midlines to the lateral alae. Based on their adult-specific expression and distribution, such puncta correspond to the cuticle struts originally observed in phase contrast and in TEM[16], and are consistent with localization of an extrachromosomal transgenic BLI-1::GFP reporter[31,33]. BLI-1::mNG animals also expressed some diffuse mNG signal in the underlying epidermis beginning in the mid-L4 stage, presumably corresponding to BLI-1::mNG in the secretory pathway. XFP knock-ins within the BLI-1 N- and C-terminal domains (*ju1789* and *ju1791* respectively) displayed similar localization (Fig. S2a, b); most analysis below focused on the *ju1789* N-terminal mNG knock-in in 1–2 day old adults.

An adult cuticle annulus typically contained three rows of BLI-1::mNG puncta: two outer rows flanking cuticle furrows (Fig. 3a, furrows indicated by yellow dashes) and a central more variable row, consistent with observations of struts using phase-contrast microscopy of isolated cuticles[16]. Peaks of BLI-1::mNG fluorescence in furrow-flanking rows were detected an average of 0.77 μm apart in the circumferential axis ($n = 10$ rows). The more variable appearance of the central row may be related to annulus width; for example, towards the tail region in hermaphrodites the annuli became wider and central row puncta were relatively large and sometimes formed two rows of small puncta. The density of BLI-1::mNG puncta was higher in the head region and anterior body with $183 \pm 10$ puncta per 100 μm² of cuticle, compared with $145 \pm 10$ puncta per 100 μm² in the middle or posterior body (Fig. 3a). BLI-1::mNG puncta were larger in the anterior body than in the midbody or posterior (Fig. 3b); in the anterior body the furrow-flanking and central row puncta were not significantly different in area. In the anterior male epidermis annuli were narrower and central row BLI-1::mNG puncta were smaller, resulting in a relatively prominent appearance of the furrow-flanking double rows (Fig. 3c); the male tail cuticle contains larger BLI-1::mNG foci that were not organized in rows (Fig. 3d). Taking the surface area of an adult *C. elegans* as approximately $5 \times 10^5$ μm², we estimate there are ~$10^6$ struts in a typical adult cuticle.

**Table 3 | Phenotypic and molecular characterization of *bli-6* alleles**

| Allele | Mutagen | Bli severity | DNA change | Predicted protein change |
|---|---|---|---|---|
| *mn4*dm | X-rays | 86% severe Bli | CGT > AGT | R71S |
| *sc16* | EMS | ND | CGT > CAT | R71H |
| *e977* | EMS | 16% Bli | CGT > CAT | R71H |
| *e519*sd | EMS | 100% severe Bli | CGT > TGT | R74C |
| *n776*sd | EMS | 30% Bli | CGT > TGT | R74C |
| *e1142* | EMS | Non-Bli, 100% nose bump | GGA > GAA | G104E |
| *ju1681* | CRISPR | 100% Tail Bli | 1160 bp deletion | Deletion of G55 to stop |

$n = 50$ scored per genotype.

Rows of BLI-1::mNG puncta were observed over almost all the lateral and dorsoventral cuticle, with mNG brightness varying depending on the region. Puncta in the head and the dorsal tail cuticle (overlying anteriormost and posteriormost epidermis hyp5 and hyp10) were brighter than in other regions (Fig. S2a, b). BLI-1::mNG puncta overlying the edges of body muscle quadrants were also brighter than those in non-muscle-adjacent epidermis, suggesting that within each epidermal cell, deposition of BLI-1 might be modulated by local mechanical forces or signals from muscle. BLI-1::mNG puncta were absent from certain regions of the cuticle, most obviously the lateral alae-generating cuticle overlying seam cells (Fig. 3a), consistent with TEM analysis showing that struts are absent from the lateral cuticle[16]. The cuticle of the male tail fan contained diffuse BLI-1::mNG (Fig. 3d); other cuticle regions containing little or no BLI-1::mNG include the anteriormost tip of the nose, the hermaphrodite vulva, and sensilla such as the postdeirids or phasmids (Fig. S2a, b). These observations suggest most BLI-1::mNG remains closely associated with the directly underlying epidermal cell.

### BLI-2 and BLI-6 localize to strut-like puncta and to other cuticle substructures

To determine the localization of BLI-2 we examined a BLI-2::mNG C-terminal knock-in (*syb3293*) (Fig. 1c). BLI-2::mNG localized to adult strut puncta in a pattern essentially indistinguishable from that of BLI-1::mNG. Additionally BLI-2::mNG localized to the lateral alae and seam-derived cuticle (Fig. 3e) and to rings surrounding sensilla (Fig. S2c). BLI-2::mNG precisely colocalized with a BLI-1::wrmScarlet marker (*juEx8105*) in strut puncta, but not in the lateral alae (Fig. 3e, Fig. S3e). As described below, in *bli-1(0)* mutants, BLI-2::mNG localization to alae was normal, but localization to strut puncta was dramatically reduced. Taken together, these observations indicate that BLI-2 colocalizes to struts, dependent on BLI-1, and independently localizes to lateral alae.

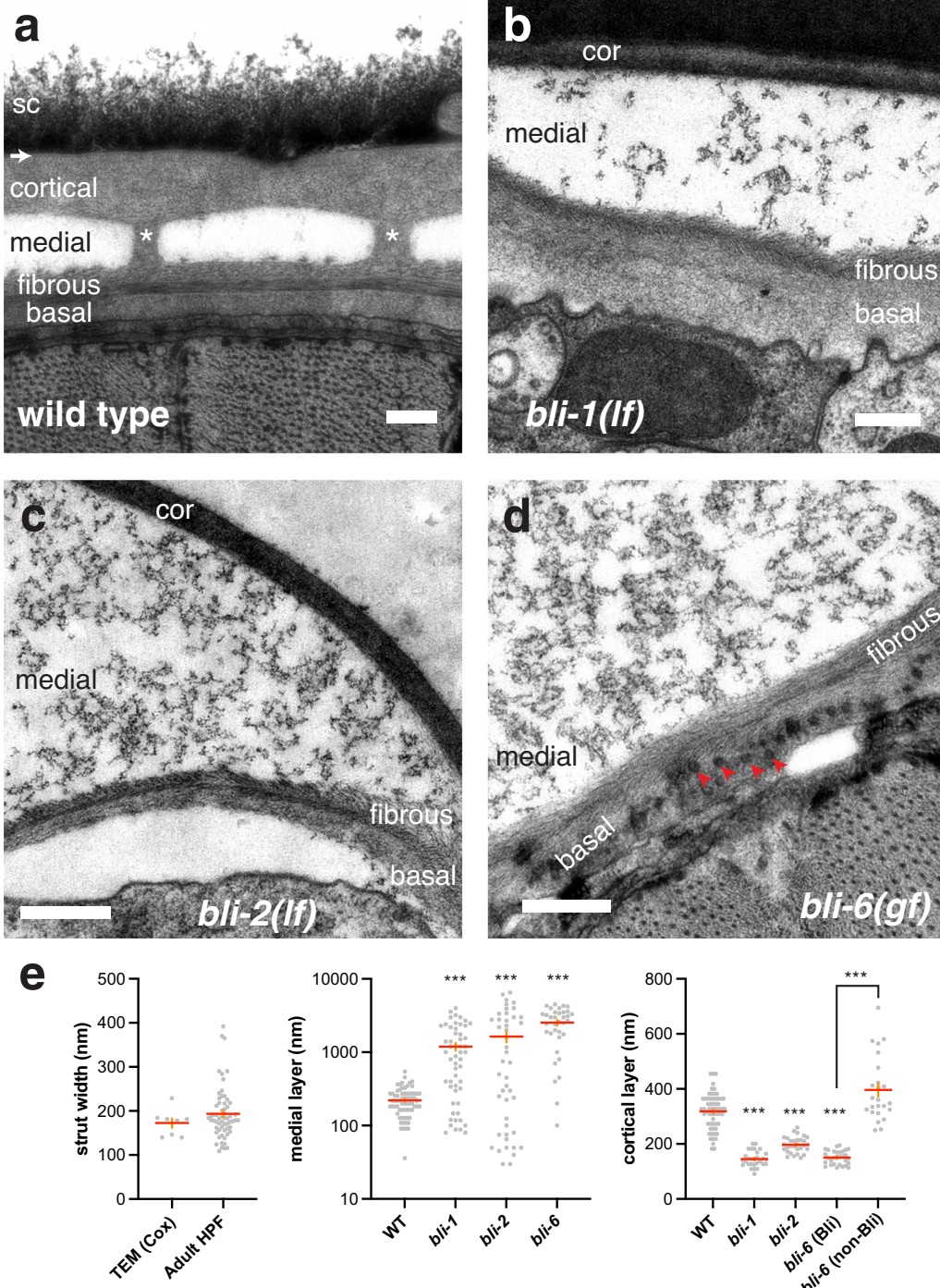

**Fig. 2 | *bli* collagen mutants are defective in strut formation.** Transmission electron micrographs (high-pressure freeze fixation; see Methods), transverse sections of cuticle of wild type and representative mutants. **a** Wild-type (N2) adult cuticle showing (from top to bottom) epicuticle (arrow), cortical (cor), medial, fibrous (fib), and basal (bas) layers. Two struts (white asterisks) connect the cortical and fibrous layers. The surface coat (sc) appeared as a variable diffuse layer external to the cortical layer. Image reference: Grid 1/Worm 3/Set 1/Image 155. **b** *bli-1(e944)* mutant cuticle with severe Bli phenotype showing expanded medial layer lacking struts, and amorphous electron-dense aggregates. The fibrous and basal layers appeared normal, with the two fibrous sublayers forming a herringbone appearance. Ref. 1/1/2/157. **c** *bli-2(e768)* (intermediate Bli) showing expanded medial layer lacking struts; in this image the fibrous layer has separated from underlying epidermis. Ref. 1/2/1/117. **d** *bli-6(n776)* (severe Bli) showing an expanded medial layer containing amorphous material; fibrous layer appeared normal; the basal layer contained numerous approximately 30–40 nm diameter vesicle-like structures (red arrowheads). Ref. 1/1/1/79. Scales, 200 nm. **e** Quantitation of strut width, medial layer thickness, and cortical layer thickness. Mean strut width in our HPF images was 193.61 ± 7.5 nm ($n$ = 61), slightly larger than calculated from in classic TEM data[15] of 172 ± 10.6 nm (mean ± SEM, $n$ = 8; $P$ = 0.37 by two-tailed Mann–Whitney test). Mean medial layer thickness in wild-type adult EM is 221 ± 11 nm (wild type, $n$ = 70) versus > 1100 nm in Bli mutants; ***$P$ < 0.0001 (exact $P$ = 0.0005 for WT vs *bli-2*), Kruskal–Wallis test and Dunn's post test. Mean cortical layer thickness was reduced compared to wild type in blistered (Bli) areas of cuticle in *bli-1(lf)*, *bli-2(lf)*, and *bli-6(gf)* mutants, and was more variable in non-Bli areas of *bli-6(gf)* mutant cuticle, suggesting reduced cortical layer thickness may be a secondary consequence of blister inflation. Ordinary one-way ANOVA and Šidák's post test; ***$P$ < 0.0001.

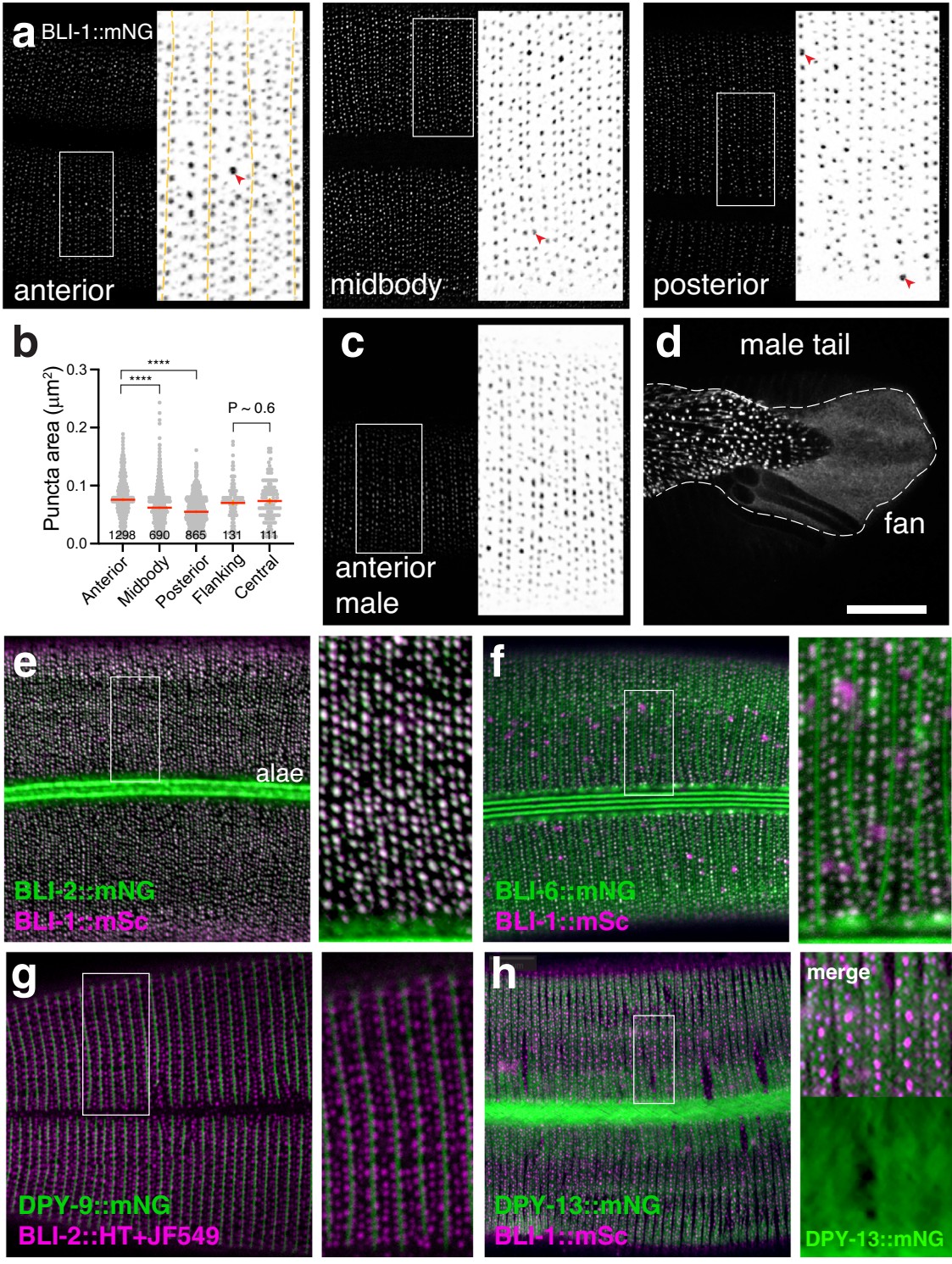

A BLI-6::mNG C-terminal knock-in (*ju1914*) localized to strut-like puncta (Fig. 3f). Unlike BLI-1::mNG and BLI-2::mNG, which tend to be more prominent in furrow-flanking strut rows, BLI-6::mNG was most prominent in the central row of struts in each annulus, confirmed by colocalization with BLI-1::wrmScarlet (Fig. 3f). Like BLI-2::mNG, BLI-6::mNG localized to adult alae, seen as three very bright parallel ridges as well as more diffuse localization adjacent to the alae proper. BLI-6::mNG additionally localized to cuticle furrows, and showed diffuse localization within the cuticle, as well as to larger foci in the deeper cuticle (Fig. 3f) as well as to rings surrounding sensilla and the vulva (Figure S2d). Analysis of confocal z-stacks suggested that the furrow-localized BLI-6::mNG signal extends through multiple planes from

cortical to basal cuticle; consistent with this, BLI-6::mNG localizes to both cortical and basal surfaces of blistered areas in *bli* mutants (see below).

## Struts localize adjacent to cuticle furrows and define regular holes in the fibrous layer

Several cuticle collagens have been shown to localize to thin circumferential regions aligned with morphological furrows[21]. We confirmed that the double rows of BLI puncta were adjacent to cuticle furrows by colocalization of BLI-2::HaloTag with a newly generated DPY-9::mNG knock-in (Fig. 3g; see also Fig. 5f). Moreover in Bli mutants the furrow collagen marker DPY-7::GFP *qxls722*[34] displayed normal

**Fig. 3 | BLI collagens localize to adult-specific puncta corresponding to struts.**
**a** BLI-1::mNG (*ju1789*) knock-in localized to puncta arranged in circumferential rows overlying dorsal and ventral epidermis but largely excluded from lateral alae. Furrows indicated by dashed yellow lines, individual puncta by red arrowheads. Larger puncta (e.g. arrowhead in left panel) tend to be more widely spaced than smaller puncta. **b** Quantitation of puncta area in BLI-1::mNG(*ju1789*). Puncta decrease in size along the body axis, comparing anterior body, midbody, and posterior body (Kruskal–Wallis test and Dunn's post test; number of puncta indicated in dot plot, taken from 3 ROIs per body region, in 3 separate experiments). In the anterior, puncta were similar in size between furrow-flanking and central rows (two-tailed Mann–Whitney test, *n* > 100 puncta per row type). Mean line in red and SEM in orange. **c, d** In adult males, BLI-1::mNG(*ju1789*) localized to regular rows of struts in the anterior body; compared to the anterior hermaphrodite, the central rows were less consistent and the furrow-flanking double rows more prominent. In the adult male tail BLI-1::mNG localized to larger puncta in the posterior tail region

and was diffusely localized in the tail fan (outlined). **e** BLI-2::mNG knock-in (*syb3293*) colocalized with BLI-1::wrmScarlet (mSc.) (*juEx8105*) (magenta) and also localized to adult lateral alae (bright green stripe). Double labeling with BLI-1::wrmScarlet and merged image. **f** BLI-6::mNG knock-in (*ju1914*) localizes to struts, alae and to cuticle furrows, and partly colocalized with BLI-1::wrmScarlet (*juEx8105*) (magenta); the *ju1914* knock-in caused mild aggregation of BLI-1::wrmScarlet and therefore may affect *bli-6* function. Conventional confocal; insets 10 μm wide.
**g** BLI-2::HaloTag puncta (labeled with JF549 staining, magenta) localized to rows adjacent to furrows marked with DPY-9::mNG (green). **h** BLI-1::wrmScarlet puncta were complementary to gaps in the fibrous layer as shown here by DPY-13::mNG knock-in (see Supplementary Fig. 3f). Top of inset shows merged channels, bottom half is DPY-13::mNG alone. Single focal plane. DPY-9::mNG stripes are in the same focal planes as BLI-1::mSc, whereas DPY-13 was more basal but partly overlapping. All images of young adult animals within 24 h of the L4/Adult molt. Scales 10 μm, insets all 10 μm wide.

localization to furrows in non-blistered areas; faint furrow-like localization could sometimes be seen on the external surface of blisters (Fig. S2a-d). Thus, BLI-1 and other BLI collagens are not essential for furrow patterning; other alterations in DPY-7 localization may be secondary to the defects in the medial layer. As shown below (Fig. 4), BLI-1::mNG in the L4 stage is initially concentrated in banded regions flanking the center of furrows, suggesting that furrows and furrow-flanking BLI-1 puncta are closely interrelated.

Other cuticle collagens such as DPY-13 localize to circumferential bands corresponding to the annuli between furrows[21]. We examined the colocalization of BLI-1 with DPY-13 using BLI-1::wrmScarlet and a DPY-13::mNG knock-in that localizes to fibrous layers underlying annuli (Fig. 3h, Fig. S3e, f). BLI-1::wrmSc puncta mostly localized to focal planes more superficial to DPY-13. Notably, in intermediate focal planes BLI-1 puncta correlated with holes or gaps in the DPY-13 fibrous layer (Fig. S3f). These observations suggest that the basal ends of struts (as defined by BLI-1::mNG) extend into fibrous layers and exclude fibrous layer collagens.

We extended these observations using the adult-specific collagen COL-19, which like DPY-13 localizes to fibrous layers underlying annuli[35]. Using a newly generated COL-19::mNG knock-in we confirmed that COL-19 localizes to fibrous layers underlying annuli (Fig. S3e, h). In *bli-1(0)* and *bli-6(gf)* blistered mutants COL-19::mNG localized to the basal but not the cortical side of blisters (Fig. S3f, g). Like DPY-13::mNG the most apical layer of COL-19::mNG shows regularly spaced holes similar in size and spacing to struts (Fig. S3e, i, j). The regularly spaced holes in the wild-type COL-19::mNG fibrous layer were absent in *bli-1* and in *bli-6(gf)* animals, although other aspects of annular or fibrous layer organization appeared normal. In *bli-6(0)* mutants, COL-19::mNG holes were altered in distribution such that holes in the center of each annulus were enlarged (Fig. S3i, j) and lateral furrow-flanking gaps were smaller, correlating with the altered distribution of BLI-1::mNG puncta in *bli-6(0)* mutants (Fig. S3h). We conclude that the holes in the superficial fibrous layer marked by COL-19 and DPY-13 correspond to the basal ends of struts, and that struts are required for formation or maintenance of these holes.

## BLI puncta condense from diffuse bands during the L4 stage
We next examined how BLI collagen puncta form during synthesis of the adult cuticle in the late L4 stage. Transcription of the three *bli* collagens is upregulated specifically in the late L4 stage, along with other adult-specific cuticle collagens[36,37]. We confirmed this temporal regulation using a *bli-1* transcriptional reporter (P*bli-1*-GFP, Fig. S4a), which displayed upregulated expression in mid-L4 stage (between 39 and 40 h after release from L1 arrest, L4.4). Upregulated P*bli-1*-GFP expression continued through the L4-Adult molt (~44 h post L1 arrest; see Methods) and then decreased to baseline levels within 24 h. These observations confirm that *bli-1* transcription is highly specific to the later L4 and early adult stage, consistent with the period of strut formation.

To understand how BLI puncta form we used timelapse imaging to examine BLI-1::mNG localization during the mid-late L4 stage (Fig. 4a, d; Supplementary Movie 1). The earliest BLI-1::mNG localization in the cuticle appeared as diffuse circumferential double bands that flanked the developing annular furrows (Fig. 4a). Faint diffuse accumulation could also be seen at the center of each annulus, corresponding to the central row of nascent puncta. Within 1 h of this initial diffuse localization, the furrow-flanking double bands of BLI-1::mNG had condensed into rows of puncta that resembled adult strut puncta; in some animals intermediate patterns could be observed, with diffuse and punctate material in the same circumferential band. Puncta in the central annular row formed at a more variable rate; some large distinct puncta could be observed at 41 h after release from L1 arrest (substage L4.6). Individual puncta appeared to form locally from diffuse BLI-1::mNG, and once formed did not further move, fuse, or separate (e.g. Fig. 4d). These observations suggest possible differences in the process of punctum formation in the furrow-flanking rows versus the central row. Each individual punctum formed over a period of 5–10 min and subsequently became brighter over ~1 h (Fig. 4d, f), but due to asynchrony the overall pattern takes > 1 h to develop under our imaging conditions. Individual puncta were variable in shape during their initial formation but subsequently became consistently circular in cross-section. In rare cases large puncta appeared to form by fusion of adjacent puncta (arrowhead, Fig. 4e). Together these observations suggest that the punctate localization of BLI-1::mNG begins within minutes after its initial secretion into the cuticle. BLI-1::mNG puncta varied in morphology during their initial formation but subsequently were stable in size or brightness.

The BLI-2::mNG signal was weaker than that of BLI-1::mNG, but generally displayed similar dynamics in the L4 stage. BLI-2::mNG initially formed patchy furrow-flanking bands in dorsolateral cuticle (~40 h post L1 arrest) which over a period of 1–2 h condensed into puncta resembling those in the mature adult cuticle (Fig. 4b; Supplementary Movie 2). Like BLI-1::mNG, BLI-2::mNG puncta in the central row did not arise from a defined band but rather formed from a diffuse pool in the annulus. In L4.3 and L4.4 stages BLI-2::mNG signal was diffusely localized in epidermal seam cells prior to its secretion into lateral alae (Fig. S4b). BLI-6::mNG showed similar dynamics, although it was more concentrated in the cuticle of the lateral epidermis in early L4 stages (Fig. 4c). In summary, the mNG-tagged BLI collagens initially localized to diffuse circumferential furrow-flanking bands as well as more diffuse material that condensed into puncta over a 1–2 h period in the late L4 stage.

We asked whether these fluorescent markers correlated with strut formation imaged using electron microscopy, taking advantage of longitudinal sections previously used in analysis of L4 vulva development[38]. Animals staged as L4.8/9[39], displayed a well-defined electron-lucent medial zone within the nascent adult cuticle (Fig. S4c). This medial zone contained diffuse electron-dense material at the base of furrows and in the middle of annuli, some of which likely

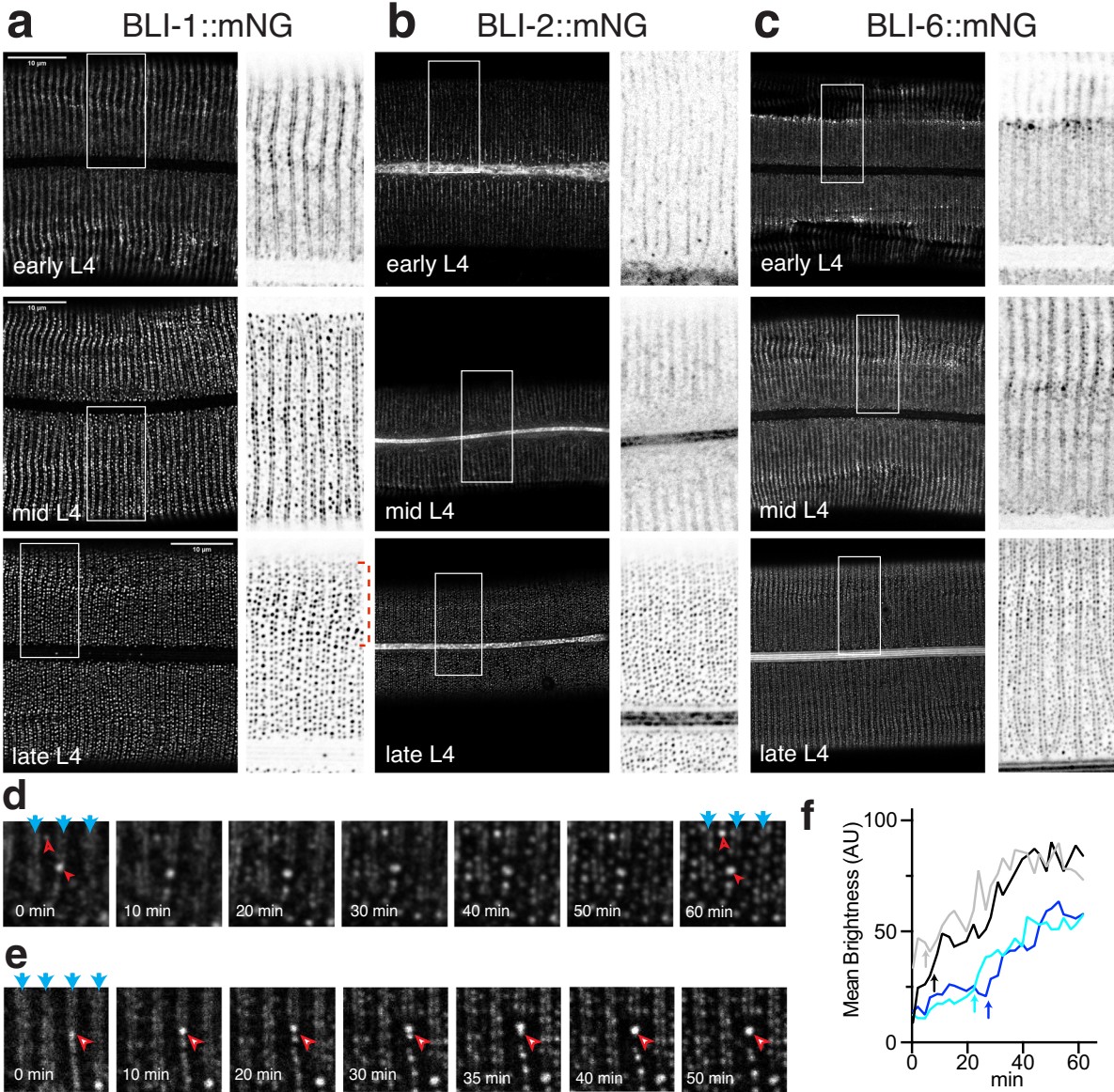

**Fig. 4 | Development of BLI puncta during L4 stage. (See also Supplementary Movies 1, 2). a** BLI-1::mNG(*ju1789*) during L4 development, lateral views of midbody. BLI-1 secretion and localization appeared to occur from L4.5-L4.8, i.e. between 5 and 8 h from the beginning of the L4 stage[39]. Beginning in L4.5, BLI-1::mNG localized to single furrow-like bands, followed by double furrow-flanking bands in L4.6, which then became progressively more punctate from L4.7–9. By late L4, BLI-1::mNG puncta in dorsoventral epidermis overlying muscle (dashed red bracket) are generally brighter than those over lateral epidermis. **b** BLI-2::mNG (*syb3293*) was first visible in diffuse bands in the middle of furrows, then at furrow-flanking bands, then condensed into puncta approximately at the same stage as BLI-1::mNG. BLI-2::mNG also localized diffusely in seam cells from L4.3 (see Fig. S4b) and subsequently in lateral alae. **c** BLI-6::mNG(*ju1914*) was generally fainter than BLI-1::mNG or BLI-2::mNG in L4 stage but also initially appeared at furrows (L4.4), then at furrow-flanking bands (L4.5–6), then in puncta (L4.6–9). In the early L4 stage BLI-6::mNG was more visible in the cuticle overlying lateral epidermis and then spread dorsally and ventrally. Alae expression of BLI-6::mNG becomes prominent at L4.7 stage. Conventional confocal, MIP of 2–4 focal planes. Scales, 10 μm; insets 10 μm wide. **d** Still images from a 25 μm² square ROI of Supplementary Movie 1 of BLI-1::mNG(*ju1789*) every 10 min. Furrow positions are indicated by blue arrows and two nascent puncta are indicated, one early-forming punctum used as fiduciary marker (solid red arrowhead) and a later-forming punctum in a central row (hollow red arrowhead). Brightness has been automatically adjusted and images smoothened from the original video. **e** Still images as in (**d**) showing a large punctum (arrowhead) forming by apparent fusion of adjacent puncta at -35 min. **f** Quantitation of brightness (arbitrary units) of individual BLI-1::mNG puncta from Supplementary Movie 1 in four representative circular ROIs (area 0.2 μm²) imaged every 2.5 min. Two early-forming puncta (black, gray) and two later-forming puncta (light, dark blue) are plotted. Overt puncta formation (arrows) corresponds to a rapid increase in mean brightness over 5–10 min followed by a slower increase over the subsequent 30–40 min.

correspond to early struts (red arrowheads). The deepest parts of the furrows appear connected to the fibrous layer (blue arrows). The cortical zone was well-formed, with an epicuticle layer (seen as two parallel electron-dense lines) lining the developing furrows. In contrast the basal zone was incompletely developed, with two fibrous layers visible but no additional basal layers. Longitudinal TEM sections of animals that appear to be earlier in the L4 stage do not show a distinct medial zone or struts[40]. In conclusion, these ultrastructural observations suggest that formation of the cortical and medial zones precedes condensation of struts.

### BLI collagens display nanoscale patterning within struts

Struts as labeled by BLI collagen fluorescent protein knock-ins appear as diffraction-limited puncta, consistent with electron microscopy data showing a typical strut diameter of 250–300 nm. To ask whether BLI collagens were patterned within struts, we turned to 3D-Structured

Illumination Microscopy (3D-SIM) which provides approximately a 12-fold improvement of volumetric resolution over conventional confocal microscopy[41]. Using 3D-SIM (see Methods), we found that BLI-1 and BLI-2 mNG knock-ins localized to cylindrical or conical structures, appearing as hollow rings in cross-section (Fig. 5a, b). Many cylinders had one or more non-fluorescent centers surrounded by a ring of fluorescence. BLI-1::mNG cylinders with non-fluorescent centers had an average peak-to-peak width of $161 \pm 39$ nm (SD, n = 100) and BLI-2::mNG cylinders displayed a width of $150 \pm 22$ nm (SD, $n = 86$) (representative line scans shown in Fig. 5c). Some struts displayed double rings or figure 8 appearances in a z-section; others appeared to be formed of partial cylinders. These cylindrical structures extended ~500 nm orthogonal to the cuticle plane, consistent with the vertical dimensions of struts in EM and confocal microscopy. Smaller struts appeared as single puncta in cross-section (e.g. the puncta in central rows in the head region). Similar patterns were observed with two BLI-1::mNG knock-ins (ju1789 and ju1791) and in BLI-2::mNG(syb3293) or BLI-2::HaloTag knock-ins (syb4687) labeled with HaloTag ligands, both in live and fixed samples, suggesting these cylindrical patterns are not dependent on the location or nature of the fluorescent protein tag (Fig. 5a). Using 3D-SIM we found that BLI-1::mNG and BLI-2::HaloTag (HT) colocalized to strut cylinders (Fig. 5d). BLI-6::mNG localization in 3D-SIM was consistent with these observations, however its diffusely localized signal precluded high-quality SIM reconstructions. Together, these observations suggest struts have a nanoscale organization that is beyond the limit of resolution of conventional confocal microscopy.

With 3D-SIM we were able to examine the relative localization of BLI collagens with respect to the furrow-associated collagen DPY-7. DPY-7::sfGFP appeared as continuous grainy bands in conventional confocal microscopy (Fig. 5e). Using 3D-SIM these DPY-7::sfGFP bands could be resolved into rows of closely-spaced puncta or short bars in individual z-sections; in the z axis these corresponded to pillars ~100–200 nm diameter extending ~500 nm orthogonally to the plane of the cuticle (Fig. 5e). Other furrow collagens such as DPY-3::mNG also showed heterogeneity within furrows, as visualized by widefield imaging and deconvolution (Fig. 5e). These observations suggest furrow collagens may also display nanoscale patterning. Colocalization of DPY-7::mKate with BLI-1::mNG using 3D-SIM revealed that furrow-flanking struts were adjacent to and in overlapping focal planes as the DPY-7 pillars (Fig. 5f). These observations suggest furrow collagens may also localize to specific regions in the medial layer.

## BLI collagens interact genetically and show interdependent localization to struts

To address whether the BLI collagens function in a common pathway in strut biogenesis we examined their genetic interactions and localization dependence. As bli-6(ju1681) null mutants displayed low penetrance tail blistering, we examined double mutants with bli-1 and bli-2 null mutants. bli-1(0) mutants typically progressed from mild Bli at 3 h post-molt to severe Bli at 9 h post-molt. In contrast, blistering in the bli-1(0); bli-6(ju1681) double mutant was intermediate by 3 h post-molt and progressed to severe by 5 h post-molt; blisters were also larger in double mutants at 24 and 48 h post-molt (Fig. 6a). bli-6(ju1681) similarly enhanced bli-2(0) in that bli-2(0) single mutants typically developed a single large head blister and several smaller blisters near the tail, whereas double mutants displayed smaller head blisters and more midbody blisters; double mutants were often decreased by 48 h after the L4/adult molt. As loss of function in bli-6 exacerbated the phenotypic severity of bli-1 or bli-2 null mutants, BLI-6 may have functions in cuticle structure independent of BLI-1 or BLI-2.

We next examined whether BLI proteins require one another for incorporation into the cuticle. In bli-2(0) mutants, very low levels of BLI-1::mNG were observed in the cuticle (Fig. 6b); scattered puncta of varying sizes could be seen in unblistered regions whereas in blisters, BLI-1::mNG localized to diffuse bands approximately the size of annuli on

the external (cortical) side of the blister and to rare scattered puncta (8.3 per 100 μm², n = 6 ROIs, compared to >100 in wild type) that were larger and brighter than wild-type BLI-1::mNG puncta (Fig. 6j). These observations suggest BLI-2 is required for regular BLI-1 punctate localization and is important in promoting BLI-1 secretion into the cuticle. Conversely, in bli-1(0) mutants BLI-2::mNG showed minimal punctate localization, but instead localized to faint circumferential bands on either side of furrows (Fig. 6e). BLI-2 localization to lateral alae appeared normal in bli-1(0) mutants. These data suggest BLI-1 is required for BLI-2 localization and organization into struts; in the absence of BLI-1, a small amount of BLI-2 is present in the cuticle but mostly remains in diffuse furrow-flanking bands. In bli-6(0) mutants, BLI-1::mNG and BLI-2::mNG localized to rows of puncta, but the distribution was altered such that the central row puncta were brighter relative to the furrow-flanking rows and larger than in the wild type (Fig. 6c, f, h). As BLI-6 itself appears enriched in the central row of puncta (see below), BLI-6 may act locally to negatively regulate BLI-1 localization. Notably, in bli-6(mn4) gain of function mutants, BLI-1::mNG and BLI-2::mNG localized to disorganized puncta of variable size in dorsoventral (muscle adjacent) regions, outside blistered areas (Fig. 6d, g, h). Thus, the gain of function BLI-6(mn4) may locally inhibit strut formation.

Unlike BLI-1::mNG and BLI-2::mNG, BLI-6::mNG was present in both cortical and basal layers in blisters (Fig. 6i). Few BLI-6::mNG puncta were visible in bli-1(0) mutants and the remaining BLI-6::mNG localized to cortical furrow-like bands over the blistered areas although some variable aggregates were also observed inside the blister (arrows, Fig. 6i). These observations suggest BLI-1 and BLI-2 promote the strut localization of BLI-6 but are not required for its localization to furrows or other cuticle substructures. In summary, BLI-1 and BLI-2 appear to promote each other's localization to puncta but are not essential for their initial secretion into the apical ECM. BLI-6 is not essential for BLI-1::mNG or BLI-2::mNG punctate localization but affects their distribution across strut rows.

To further understand the biochemical relationship between the BLI collagens we examined their maturation using Western blots of preparations of soluble cuticle protein (see Methods). The mature adult cuticle displays extensive crosslinking[42] so we focused on the nascent adult cuticle during L4 stage (Fig. S5). BLI-1::mNG was detected beginning in early L4 stage, in 3 major bands consistent with processed monomeric (125 kDa), homodimeric (250 kDa), and homotrimeric (>250 kDa) forms; an additional band of ~150 kDa of unknown identity was frequently observed. Bands consistent with monomeric BLI-2::mNG and BLI-6::mNG (predicted m.w. ~60 kDa) were consistently detected in mid-L4 stages and later, with additional bands consistent with dimeric and higher molecular weight (HMW) forms present in later L4s (Fig. S5a). BLI-6::mNG displayed apparent maturation from monomeric to dimeric or HMW forms during late L4. These observations suggest puncta formation is not a simple consequence of BLI collagen crosslinking or multimerization, and that BLI-6 may mature later than BLI-1 and BLI-2. We next tested whether altered processing of BLI proteins could be detected in the subtilisin cleavage site mutants bli-6(mn4 R71S) and bli-1(ju1972 R75C). In bli-6(mn4) mutants a larger BLI-6::mNG band of ~75 kDa was observed consistent with a partial block of cleavage of the N-terminus (Fig. S5b). In contrast the cleavage site mutation in bli-1(ju1972) resulted in very little BLI-1::mNG in the soluble cuticle fraction, suggesting cleavage of BLI-1 may be important for its secretion. Finally, we examined the molecular epistasis of BLI-1 and BLI-2. In bli-2(0) mutants a reduced amount of BLI-1::mNG was detected in the cuticle fraction, consistent with BLI-2 being important for BLI-1 secretion (Fig. S5c). Interestingly, bli-1(0) cuticle fractions displayed a reduced amount of BLI-2::mNG in HMW forms and increased levels of monomeric or dimeric forms, suggesting BLI-1 may be required for BLI-2 to form higher order multimers. These findings are also consistent with imaging showing that BLI-1 and BLI-2 are partly or variably required for each other's accumulation in the cuticle (Fig. 6b, e).

### BLI collagens interact with furrow and annular collagens

The cuticle is patterned by complex interactions between the underlying epidermal cytoskeleton and by interactions between the different cuticle substructures such as annuli and furrows. Moreover, Bli phenotypes can be suppressed in double mutant combinations with other cuticle mutants, some of which also display altered strut distribution by phase-contrast microscopy[43]. To better understand the interactions between struts and other cuticle structures we surveyed a

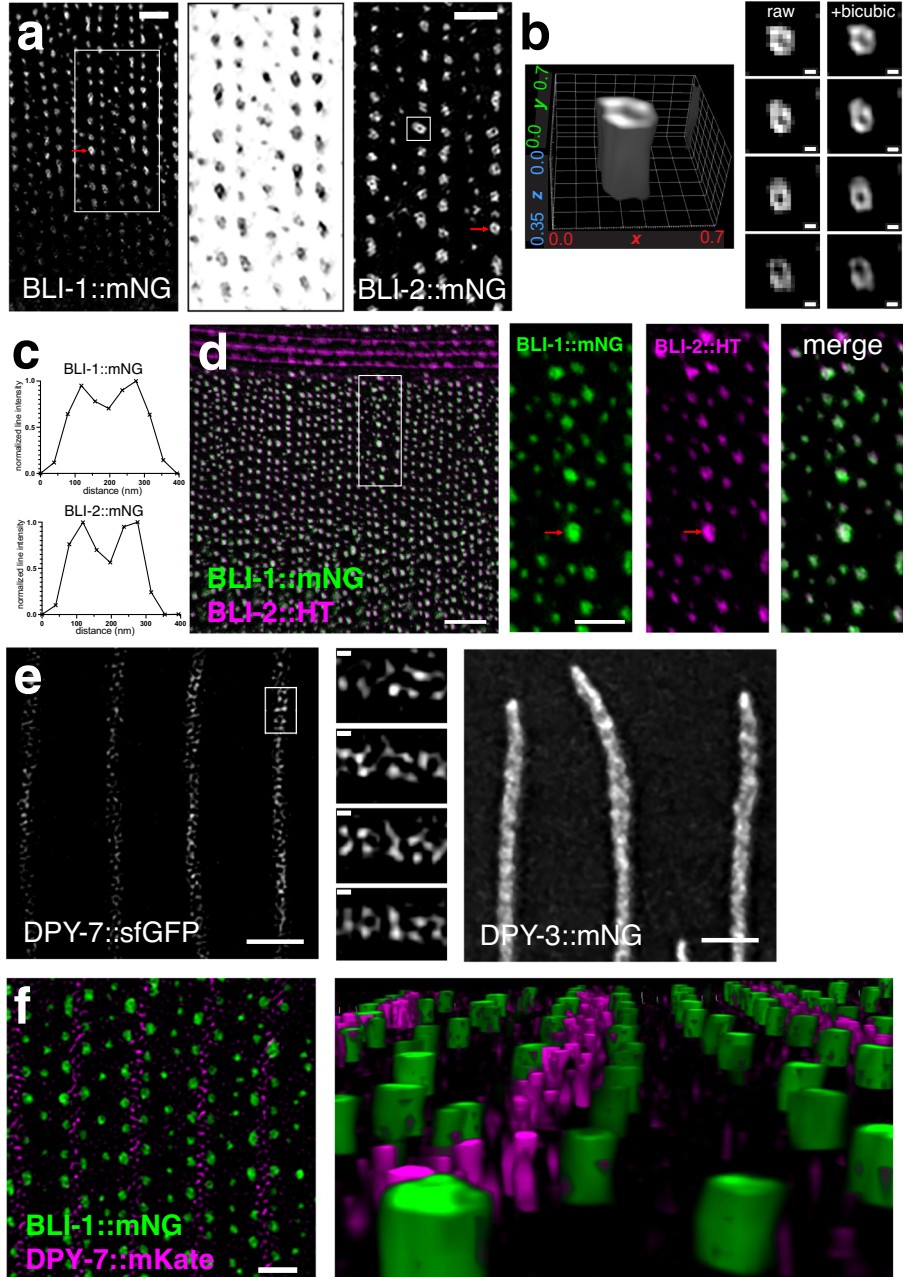

**Fig. 5 | Nanoscale organization of BLI collagens. a** Cylindrical architecture of struts in 3D-SIM imaging as visualized by BLI-1::mNG(*ju1789*) and BLI-2::mNG(*syb3293*). Single focal planes; BLI-1::mNG inset 4 μm wide, inverted grayscale. Scales, 1 μm. Average feature MCNR = 12.2 (BLI-1::mNG) and 13.5 (BLI-2::mNG). **b** 3D rendering and enlargement of single BLI-2::mNG strut (boxed in panel **a**) in 4 z-sections (125 nm steps), before and after bicubic interpolation (scales: 100 nm). Grid lines in rendering are spaced at 0.1 μm intervals in x and y axis and 0.05 μm in z. **c** Representative line scans (subpixel point-to-point measurement, Softworx) across single BLI-1::mNG and BLI-2::mNG struts (marked with red arrows in **a**). Mean peak-to-peak diameter for struts exhibiting non-fluorescence centers was 161 ± 29 nm for BLI-1::mNG (SD, *n* = 110 struts) and 150 nm ± 22 nm for BLI-2::mNG (*n* = 86). **d** Precise colocalization of BLI-1::mNG (green) and BLI-2::Halo-Tag(*syb4687*) stained with JF549 (magenta) in 3D-SIM. Insets (boxed) are 4 μm wide, Scale 4 μm or 2 μm in inset. Average feature MCNR = 8.2 (BLI-1::mNG), 9.1 (BLI-

2::HT). Note that the non-fluorescent strut centers are less evident for BLI-2::mNG due to lower resolution at 588 nm. In some cases BLI-2::HT localized to smaller structures than the BLI-1::mNG rings (red arrows), consistent with the slightly smaller BLI-2::mNG ring width compared to BLI-1::mNG. **e** Columnar organization of furrow-associated collagens. DPY-7::sfGFP(*qxIs722*) in furrow regions as visualized in 3D-SIM. Single focal plane (scale 2 μm) and 4 × 125 nm z planes (ROI boxed in left-hand panel), scale 100 nm. Average feature MCNR = 7.3. Comparison image (MIP of z-stack of 8 × 125 nm planes) of DPY-3::mNG showing heterogeneity within furrows, as imaged with OMX widefield and deconvolution; scale 2 μm. **f** BLI-1::mNG (green) and DPY-7::mKate (magenta) localization in overlapping focal planes of medial layer (cf. Fig. 3g). Single focal plane from 3D-SIM (left) and 3D rendering of DPY-7 columns (magenta) and BLI-1::mNG struts (green). Scale, 2 μm. Average feature MCNR = 11.7 (BLI-1::mNG) and 4.0 (DPY-7::mKate). All images are of young adult animals within 24 h of the L4/Adult molt.

subset of molecularly defined collagen mutants (Table S1) for interactions with *bli-1* and *bli-2* null mutants, and effects on BLI-1 localization. Many mutants causing short, fat Dumpy body morphology affect cuticle collagens[18], including a subset of 6 that localize to and/or function in formation of furrows (DPY-2,-3,-7,-8,-9, -10) and a pair of

collagens localized to annuli (DPY-5, DPY-13)[21]. Loss of function mutations in such furrow or annular DPY collagens strongly suppressed Bli phenotypes of *bli-1(0)* and of *bli-2(0)*, such that double mutants were Dpy but not Bli (Fig. 7a; Table S1). Mutations in the cuticle collagens DPY-4 and DPY-17 showed divergent effects: *dpy-4(lf)*

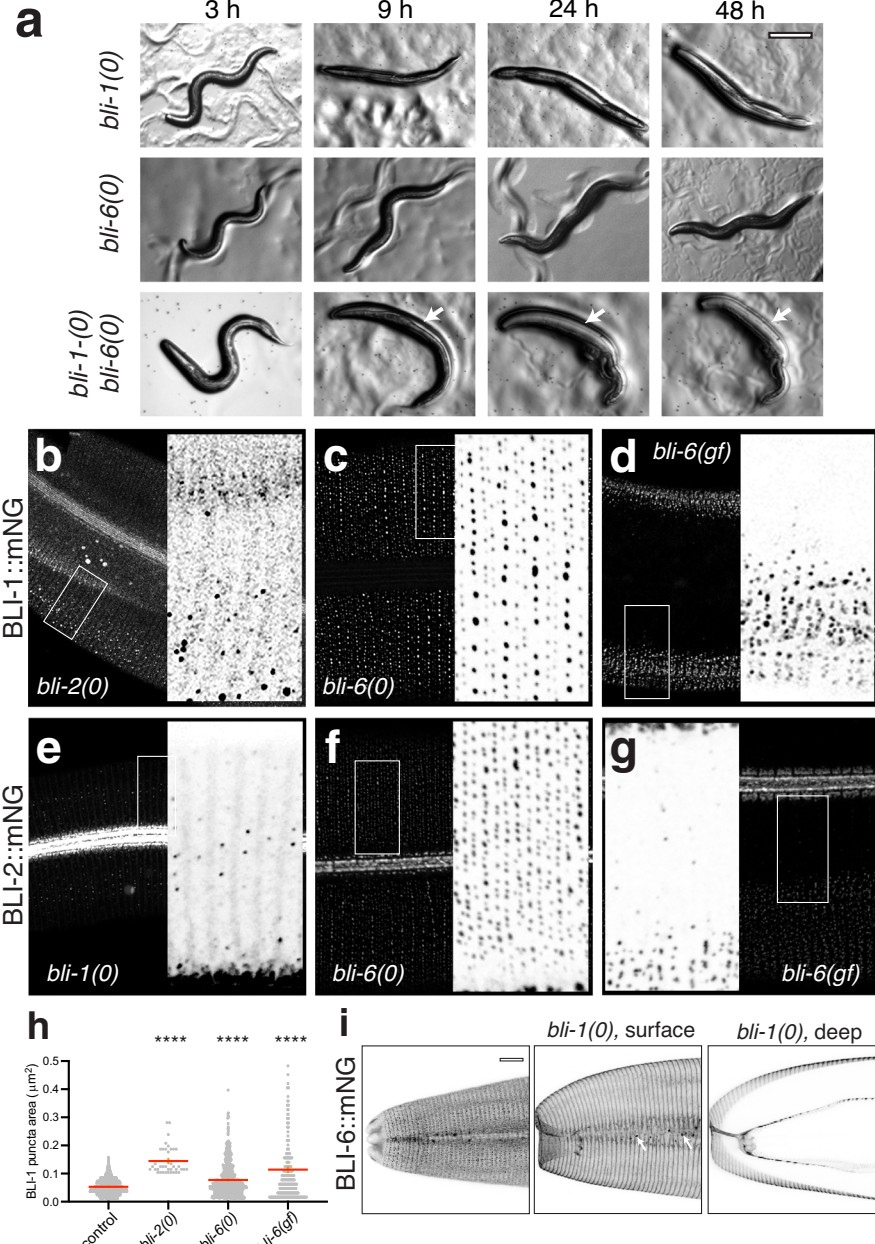

**Fig. 6 | Interactions between BLI collagens. a** *bli-6(ju1681)* enhances blistering in *bli-1(0)* mutants. Dissection scope images of representative animals between 3 and 48 h post L4/adult molt. *bli-1(0)* mutants display small blisters (arrows) within 3–5 h of the L4/adult molt and are severely blistered by 9 h. Double mutants display more widespread blisters by 9 h (arrows) and are often dead by 48 h. Similar enhancement was observed between *bli-2(0)* and *bli-6(0)*; n = 10 animals observed longitudinally per genotype. Scale, 250 μm. **b** BLI-1::mNG(*ju1789*) puncta are mostly absent in *bli-2(0)*, but occasional sparse and randomly organized puncta are seen in some regions. Confocal image enhanced relative to wild type to detect fainter puncta. **c** In *bli-6(ju1681)*, BLI-1::mNG(*ju1789*) puncta are aberrantly distributed such that the central row puncta are brighter relative to furrow-flanking rows. **d** In *bli-6(mn4)*, BLI-1::mNG(*ju1789*) puncta are disorganized and restricted to regions overlying body wall muscles. **e** In *bli-1(0)* mutants, BLI-2::mNG localizes to faint bands on either side of blistered areas and to sparse puncta in the central annular

row. **f** In *bli-6(0)* null mutants the distribution of BLI-2::mNG is altered such that the central row is stronger than lateral (furrow-flanking) rows. **g** In *bli-6(mn4)* gain of function mutants, BLI-2::mNG puncta are reduced and disorganized and restricted to areas of cuticle overlying muscle. Exposures of mutant images are enhanced compared with wild type. **h** Quantitation of BLI-1::mNG puncta size in *bli-2(0)*, *bli-6(0)* and *bli-6(gf)* mutants. ****$P < 0.0001$, Kruskal–Wallis test and Dunn's post test; error bars (orange) are SEM. **i** In the wild-type BLI-6::mNG(*ju1914*) localizes to struts and furrows; maximum intensity projection of head cuticle. In *bli-1(0)* mutants BLI-6::mNG puncta do not form; the remaining BLI-6::mNG localizes to furrow-like bands on the outer (cortical) and inner (basal) surfaces of blisters. Maximum intensity projections of 5–10 surface or deep confocal planes of BLI-6::mNG(*ju1914*); *bli-1(0)*. Scales 10 μm, insets 10 μm wide. All images are of young adult animals within 24 h of the L4/Adult molt.

caused strong suppression, whereas *dpy-17(lf) bli* double mutants were not suppressed, being Dpy and Bli. Loss of function in the LON-3 collagen[44] did not suppress Bli phenotypes, instead, Lon phenotypes were suppressed such that double mutants were Bli non-Lon (Fig. 7a). These observations suggest struts may have functions related to body length, in addition to cuticle integrity. Gain of function mutations in two collagens SQT-2 and SQT-3 have complex effects on cuticle morphology and showed strong suppression of the Bli phenotype (Fig. 7a). This limited survey indicates complex genetic interactions between struts and other cuticle components. Moreover, suppression of the Bli phenotype does not appear to be a straightforward consequence of altered body morphology, nor is it clearly related to the specific localization of the interacting protein. We further examined BLI-2::mNG in *bli-1(0) dpy* or *sqt* double mutants and found that strut number and morphology was not restored, although occasional scattered puncta were visible as in *bli-1(0)* single mutants (Fig. 7b). These observations suggest phenotypic suppression of the Bli phenotype can result from bypass of the requirement for struts in the adult cuticle.

We further tested whether these cuticle collagen mutants affected BLI-1::mNG localization. Mutations in furrow-localized collagens, such as *dpy-3(lf)*, *dpy-8(lf)* or *dpy-9(lf)*, cause annuli to be fragmented and disorganized[11,35,40]. We found that such mutants displayed aberrant and fragmented strut rows that were misoriented in random directions (Fig. 7c; Table S1). BLI-1::mNG puncta in these aberrant rows were larger and sparser than in the wild type (Fig. 7D); ectopic BLI-1::mNG puncta were also seen in regions of cuticle beneath alae, for example in *dpy-3* (arrowhead). In contrast, loss of function in annular collagens (*dpy-5, dpy-13*), which result in mild disruption of furrows, had less dramatic effects on BLI-1::mNG localization although some disorientation of rows was apparent (Fig. 7c). Loss of function in the *lon-3* collagen causes widening of annuli;[35] in such mutants BLI-1::mNG puncta rows were normal in orientation but individual puncta were larger or brighter than in wild type. Gain of function mutations in *sqt-2* and *sqt-3* cause severe disruption to the pattern of the annular collagen COL-19[35] and have been reported to lack struts[43]. We found that *sqt-2(gf)* and *sqt-3(gf)* mutants displayed widespread disruption and disorganization of strut rows, however BLI-1::mNG puncta were present (Fig. 7c). Taken together these results are consistent with cuticle furrows playing a key role in strut organization and patterning.

## Discussion

The formation of the intricate architecture of apical extracellular matrices remains poorly understood in many organisms. Here we focus on *C. elegans* cuticle struts, which display multiple levels of spatial patterning. We show that three collagens BLI-1, BLI-2, and BLI-6, play critical roles in strut formation. Loss of function in BLI-1 or BLI-2 specifically affects struts, whereas BLI-6 plays a distinct role as discussed below. Fluorescent protein knock-ins to endogenous loci reveal that BLI-1, BLI-2, and BLI-6 localize to struts: BLI-2 and BLI-6 also localize to adult alae, and BLI-6 additionally localizes to furrows. Our timelapse analysis has revealed the timecourse of biogenesis of strut puncta. Using 3D-SIM super-resolution microscopy we have uncovered previously unknown nanoscale organization of struts. Our studies establish strut formation as a tractable model of aECM patterning.

### Functional analysis of BLI collagens

Mutations causing Bli phenotypes have been isolated in numerous phenotypic screens and define only six genes, three of which encode collagens. Extensive screens have not yet identified other *bli* loci suggesting a limited number of proteins may have specific and non-redundant roles in adult strut formation. Our molecular genetic analysis of *bli-1* and *bli-2* establishes their null phenotypes as severe adult-onset blistering. In contrast null mutants in *bli-6* cause subtle effects on cuticle morphology and accelerate the onset of blistering in *bli-1* or *bli-2* null mutants, suggesting BLI-6 may function separately

from BLI-1 or BLI-2. Instead, severe *bli-6* mutations are due to Arg substitutions in the subtilisin cleavage site, similar to dominant gain of function mutations in other cuticle collagens such as *sqt-1*[29] that likely block subtilisin cleavage[42]. The presence of uncleaved forms of BLI-6 might affect the processing or assembly of BLI-1 or BLI-2, or of additional proteins involved in strut formation.

The *C. elegans* genome contains ~178 cuticle collagen genes, classified according to sequence features such as the arrangement of Cys residues[18] or patterns of inserts within the Gly-X-Y domain[17]. BLI-2 and BLI-6 are typical for nematode cuticle collagens in size, whereas BLI-1 is the largest cuticle collagen in *C. elegans* due to its extended N-terminal and C-terminal regions. Our mutational analysis suggests the extended N-terminus is important for BLI-1 function. Speculatively, the N and C-terminal regions could be involved in anchoring the BLI-1 collagen to the cortical and basal layers. Although we have not observed differences in the localization of the N-terminal and central BLI-1 knock-ins, this possibility could be investigated with additional knock-ins and super-resolution imaging.

BLI-1-like collagens with extended N- and C-terminal domains can be identified in the genomes of multiple species in the *Caenorhabditis* genus[45], with highest similarity to BLI-1 within the Elegans supergroup. Genomes of other members of the genus such as *C. bovis* contain more distantly related potential BLI-1 proteins with shorter N- and C- termini, suggesting the extended BLI-1 N-and C-termini may have evolved within the *Caenorhabditis* genus. Searches of non-*Caenorhabditis* nematode genomes have not yet revealed obvious BLI-1 orthologs, however as nematode cuticle collagens are multigene families defined by simple sequence repeats, further investigation is needed to fully understand the molecular evolution of strut collagens and how they correlate with cuticle morphology.

### Localization and interdependence of BLI collagens

We find BLI-1, BLI-2, and BLI-6 tagged proteins all localize to struts. Within the limits of our imaging resolution these proteins appear to label similar structures, although we note that the BLI-1 cylinders appear slightly wider than BLI-2 cylinders in our 3D-SIM analysis. Further analysis will be required to determine the precise spatial relationship of these proteins within struts. We find BLI-1 and BLI-2 are largely required for each other to localize to strut puncta; in the absence of one, the other is mostly diffusely localized, suggesting puncta formation requires both BLI-1 and BLI-2. As a few aberrant puncta can be detected in *bli-1(0)* or *bli-2(0)* mutants, each protein may have limited capacity to form puncta in the other's absence. The similarity in the genetics and localization of BLI-1 and BLI-2 raises the question of whether they form a heteromultimeric complex. In vertebrates, collagen heterotrimeric partners can affect each other's secretion;[46] we find that BLI-1 secretion into the cuticle or punctum formation is dramatically reduced in the absence of BLI-2, and vice versa. However our biochemical analysis has not revealed clear-cut evidence for heteromultimers; instead, the apparent molecular weights of BLI::mNG proteins is generally consistent with homodimers or multimers. BLI-1 and BLI-2 could form independent oligomers that subsequently interact to form higher order structures, as has been hypothesized for the fibrous layer collagens DPY-17 and SQT-3[47].

Complete loss of function in *bli-6* causes mild Bli phenotypes and alters BLI-1::mNG distribution such that the furrow-flanking puncta are smaller and the central annular row puncta were larger and brighter. BLI-6 appears to localize preferentially to the central row of struts versus furrow-flanking puncta, suggesting that BLI-1 may redistribute between struts to compensate for lack of BLI-6. BLI-6 is also the only collagen currently known to localize to both struts and furrows, suggesting these cuticle substructures may be related in composition and patterning. Further investigation will be needed to determine the exact role of BLI-6 in strut and furrow formation.

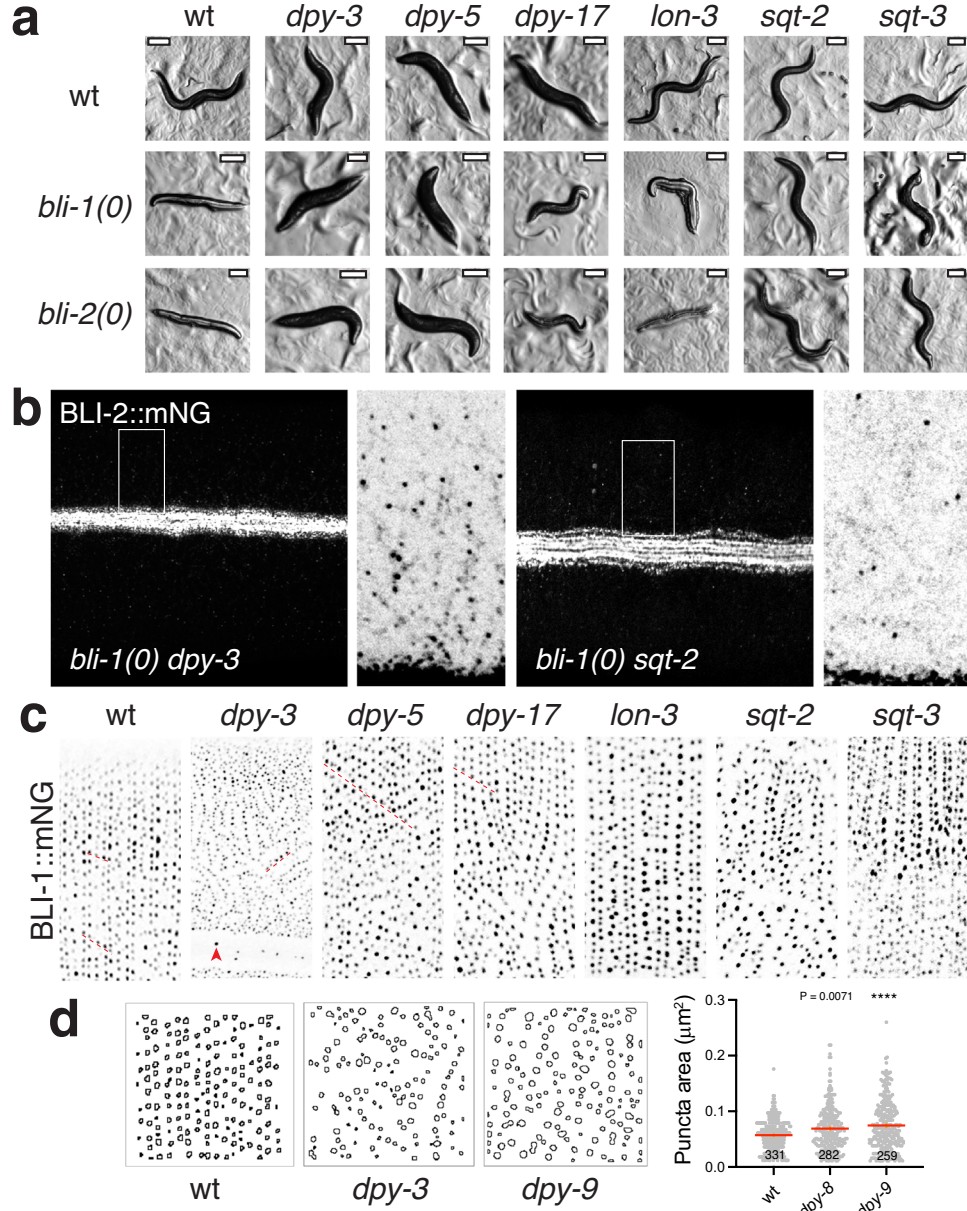

**Fig. 7 | Genetic interactions between *bli* and other cuticle collagen mutants.**
**a** Phenotypic interactions between cuticle morphology mutants (see Table S1). *dpy-3* and *dpy-5* cuticle morphological phenotypes (Dpy) are epistatic to *bli-1* and *bli-2* null phenotypes such that double mutants are Dpy non-Bli. *dpy-17 bli* double mutants are severely Dpy and Bli. *bli lon-3* double mutants are normal in length and severely Bli. *sqt-2* double mutants strongly suppress Bli; *sqt-3*(ts) double mutants at 20 °C (shown) display partial suppression of Bli. Scales, 200 μm. **b** Suppression of Bli phenotypes of *bli-1(0)* by *dpy* or *sqt* mutants does not restore BLI-2::mNG(*syb3293*) puncta. In non-Bli (suppressed) *bli dpy-3* or *bli sqt-2* double mutants, randomly scattered puncta are visible, similar to BLI-2::mNG in *bli-1(0)* single mutants (Fig. 6e). BLI-2::mNG localization to alae is normal. Boxes and insets 4 μm wide. **c** BLI-1::mNG localization in cuticle collagen mutants; representative images of BLI-1::mNG(*ju1789*). Longitudinal chains of puncta are indicated by red dashed lines. In mutants defective in furrow collagens such as *dpy-3(e182)*, BLI-1 puncta form short rows in variable orientations in the cuticle overlying the lateral epidermal ridge; the cuticle overlying dorsoventral body wall muscle quadrants displays more extensive disorganization. The shorter puncta rows vary in longitudinal spacing and often branch or join. A few ectopic scattered puncta are seen in the lateral alae region (red arrowhead). In mutants such as the annulus collagen *dpy-5(e61)*, circumferential BLI-1::mNG rows are mostly properly oriented but dorsoventral regions display chains of puncta aligned at an angle to the longitudinal axis (underlined with red dashed line). *dpy-17* mutants display normal puncta size and distribution with some aberrantly oriented circumferential rows. *lon-3* mutants display brighter puncta especially in central rows, correlating with increased annulus width. *sqt-2(sc3)* and *sqt-3(sc63)* mutants show widespread disorganization that is more severe laterally; partial circumferential rows are visible (e.g. in *sqt-3*). Maximum intensity projections of z-stacks, 10 μm wide. **d** BLI-1::mNG puncta are enlarged in furrow-defective mutants *dpy-8* or *dpy-9*. Puncta measured in midbody as in Fig. 3b; n shown on bar chart. Statistics, Kruskal–Wallis test and Dunn's post test, ****$P < 0.0001$; mean and SEM indicated. Images are of young adult animals within 24 h of the L4/Adult molt.

Our analysis has revealed aspects of strut architecture that were not previously evident from conventional light microscopy or from EM studies. For example, although struts appear amorphous and continuous with cortical and fibrous layers in TEM, confocal and 3D-SIM imaging of individual strut components such as BLI-1 indicates that struts extend into the fibrous layer and define regularly spaced holes in the fibrous layer. Mutants lacking struts also lack fibrous layer holes, suggesting struts pattern the fibrous layer. Moreover, using super-resolution imaging we find struts display internal organization into tube or cone shaped structures. As struts appear as solid

electron-dense columns in TEM, the centers of struts may contain other as-yet unidentified proteins.

## Multiple levels of patterning of BLI collagens and struts

The first level of strut patterning is formation of circumferential rows along most of the cuticle. The regular spacing of struts in a circumferential row does not appear to reflect any underlying epidermal cytoskeletal periodicity, suggesting it is generated by interactions between proteins in the cuticle. In addition, puncta are not randomly positioned with respect to adjacent circumferential rows but can form short straight or curved chains. Circumferential spacing is similar in cuticle overlying the dorsoventral (muscle-quadrant) or lateral (ridge) parts of the hyp7 epidermal syncytium. However the cuticle regions overlying the lateral epidermal ridges appear more sensitive to disruption in cuticle mutants, a phenomenon that has also been seen in localization of the furrow collagen DPY-7[34]. The presence of muscle-epidermal attachment structures in the dorsoventral epidermis may serve to reinforce strut positioning even in the absence of some cuticle components.

Strut rows themselves are patterned into central and furrow-flanking rows. Furrow-flanking rows are the most consistent in puncta size and spacing and are prefigured as diffuse bands in the L4 stage cuticle furrows, suggesting furrows are important in organization and/ or secretion of strut material. Each annulus also contains a central row or rows of puncta that are more variable in size. The variability of the central strut rows had been observed in phase-contrast microscopy[16] and may be a function of annulus width (i.e. the total amount of local collagen) in that the central row appears more prominent or duplicated in wider annuli (e.g. in lon-3 mutants). Unlike the furrow-flanking rows the central row is not prefigured by a diffuse band in the L4, suggesting possible differences in the mechanism of punctum formation between central and furrow-flanking rows.

BLI::mNG puncta are often brighter in areas overlying body wall muscle boundaries, suggesting strut formation may be modulated by local mechanical forces. Struts in regions overlying tail epidermal cells such as hyp10, or the male tail, are larger and appear more randomly distributed. Conversely, BLI-1::mNG puncta are absent from cuticle regions overlying specialized epithelial cells (seam, tip of head, vulva, excretory pore) suggesting that strut puncta do not move laterally beyond the boundaries of the underlying epidermal cell. The cuticle of such interfacial epidermal cells is thought to be unlayered[31]. Unlike BLI-1 and BLI-2, BLI-6 is observed in vulval cuticle, and its function in the vulva remains a topic for future investigation. Thus of the three collagens localizing to struts, only BLI-1 appears specific to struts whereas BLI-2 and BLI-6 localize to additional matrix substructures independent of BLI-1. These observations suggest ECM components can form highly specific structures despite not being exclusively localized to such structures. Additional questions for future analysis will be the nature of the epidermal or muscle signals that modulate strut size or intensity. In summary, these observations indicate that struts are patterned by the integration of a complex set of cues, including signals from underlying cells and interactions within the cuticle itself.

## Strut biogenesis in the L4 stage

The initial localization of BLI-1, BLI-2, and BLI-6 fusion protein reporters suggests that strut components are initially secreted by regions of the epidermis underlying or adjacent to furrows. Classic ultrastructural studies of the embryonic and early larval C. elegans cuticles indicated that epidermal ridges corresponding to annuli were a major site of cuticle secretion[48]. In contrast, our observations of the dynamics of cuticle secretion in the L4 suggest furrow regions may be sites of initial secretion. Whether or not the epidermis underlying furrow regions is the main site of collagen secretion for the adult cuticle, furrows appear to play important roles in organization of other cuticle components. Prior ultrastructural and light microscopy data had suggested that the

struts in rows flanking furrows were more consistent in size and spacing than struts in the centers of annuli. Our analysis of knock-in markers confirms this pattern and has also revealed that central rows of BLI-1 or BLI-2 puncta are not preceded by overt circumferential bands other than diffuse material throughout the developing annulus. Intriguingly, bli-6(0) mutants display enhanced localization of BLI-1::mNG at the central strut row. Taken together these observations suggest the central and furrow-flanking rows of struts may form in distinct ways.

Our analysis of bli-1 transcriptional reporters indicates that bli gene transcription is under precise temporal control in the L4 stage, consistent with transcriptomic studies[20,36]. bli-1 is one of several collagen genes whose transcription is coordinately upregulated in the mid-L4 stage by the transcription factor LIN-29[19,49]. Speculatively, such adult-specific cuticle components could be additional components of struts or factors involved in strut biogenesis.

Our timelapse analysis of BLI::mNG localization in the L4 stage has begun to reveal the outlines of strut biogenesis. Initially, puncta appear to form by condensation or accretion of local diffuse material over the course of a few minutes, and subsequently increase in size or brightness over 1–2 h. Interestingly, larger puncta tend to be more separated from other puncta, suggesting that each punctum is determined by the local level of strut components. The initial punctum formation process displays visual similarities to liquid droplet formation by phase separation[50], and it is notable that some components of the C. elegans pharynx cuticle are also predicted to undergo phase separation[51]. Although mature struts do not appear to display behaviors such as fission or fusion, it is possible that nascent puncta undergo a liquid-like phase that then matures into a more stable form resulting in puncta of consistent size, analogous to that observed with droplet formation by the ECM protein tropoelastin[52]. Our biochemical data have not shown clear evidence of maturation of strut components such as a transition from monomeric to higher molecular weight forms, however our analyses are restricted to the soluble cuticle fraction and so may not address insoluble matrix structures. Further investigation will be needed to address the relevance of phase separation to strut formation.

## Genetic interactions among collagens

Mutations in multiple cuticle collagens, including those defined by morphological roller phenotypes, have long been known to suppress Bli phenotypes[43]. Here we have extended these observations using bli-1 and bli-2 null mutants and other collagen mutants whose molecular basis is now known. We find that suppression of Bli phenotypes is not simply due to altered morphology, as dpy-17 mutants do not suppress Bli phenotypes. Moreover, suppressed bli-1 dpy-3 or bli-1 sqt-2 double mutants do not restore strut puncta (defined by BLI-2::mNG) either in numbers or pattern. Instead, the suppression of blistering may reflect failure to form a medial layer, or aberrant adhesion of cortical and basal layers. In wild-type adults, the seam-derived cuticle beneath the lateral alae lacks a medial zone or struts and instead the cortical and basal layers are closely apposed. Suppression of blister formation might also reflect the default formation of cuticle more resembling those of larval stages, which do not contain a medial layer or struts. As some suppressed double mutants such as bli-1 sqt-2 display relatively normal body length, these results suggest struts are not essential for normal adult morphology.

Early studies reported that cuticle roller mutants such as sqt-2(sc3) and sqt-3(sc63) lacked struts as visualized by phase-contrast microscopy[43]. We re-examined some of these mutants and found that they display abundant BLI-1 puncta of approximately normal size, though with aberrant global patterning. The basis for this possible discrepancy with prior observations from phase-contrast microscopy remains to be determined; as yet, we have not found any mutations that eliminate formation of struts, other than in the three bli collagens.

## Role of struts and the medial layer

Why do adult *C. elegans* cuticles contain struts or a medial layer, given that larval cuticles do not contain these structures? Although our study does not directly answer this question, it seems probable that struts play multiple roles in the structural and functional integrity of the mature cuticle. Notably, a fluid-filled median or medial layer or zone is found in many nematode cuticles and may have arisen multiple times in nematode evolution[53]. Such fluid-filled layers can contain simple pillar-like struts, as in *C. elegans*, but can also contain structures of varying morphological complexity ranging from "amorphous accumulations of electron-dense material"[54], complexes of rods and rings[55] up to elaborate systems of plates and fibers[22]. In many species these structures appear to provide mechanical linkage between the inner and outer cuticles, although in some (e.g., *N. brasiliensis*) struts appear to be suspended in the medial zone by fibrils[56]. The genetic analysis of *C. elegans* BLI collagens indicates struts primarily provide a strong mechanical connection between the adult cuticle layers, as well as potentially playing roles in sensing damage or alterations to cuticle integrity. In addition, *bli-1* mutants are defective in the response to auditory stimuli[13], suggesting struts might contribute to mechanosensation.

A related question is why defects in struts should result in dramatic expansion of the medial layer leading to overt blister formation. The role and composition of the medial layer is not well understood, but its function as a fluid compartment may be important for aspects of cuticle function. Recent studies have identified the GM2AP-related lipid transfer protein GMAP-1 that accumulates in blisters in *bli-1* mutants and which contributes to permeability barrier function[57]. Further, loss of function in GMAP-1 reduces blistering in *bli-1* mutants, suggesting blistering might result from defective trafficking of lipids or other components to the cuticle surface. The medial layer in general might play a role in lipid transport in the cuticle, and struts could contribute to cuticle lipid homeostasis. Interestingly in some parasitic species such as *Trichuris*, the cuticle can display localized blister-like inflations[58,59] implicated in attachment to host epithelia. Such phenomena suggest that blistering or inflation of the cuticle, possibly by modulation of struts or their equivalents, could be functionally important in nematodes adaptation to their environment.

In conclusion, we have revealed new aspects of the patterning of the *C. elegans* apical extracellular matrix in vivo. Our molecular genetic analysis of struts could stimulate avenues for investigation of other apical ECMs that display nanoscale architecture or collagen-based suprastructures[60]. In *C. elegans*, the aECM of the pharynx and buccal capsule contains several nanoscale structural specializations such as sieve extensions or circumferential ribs, likely composed of chitin and pharynx-specific proteins; the mechanisms by which these are patterned is not yet understood[51]. More generally, complex nanoscale patterns in the extracellular matrix have been observed in many organisms, including in insect cuticle[6], in the external tunic of *Oikopleura*[61], in collagenous endomysial struts of vertebrate heart muscle[62], and in the tectorial membrane of the mammalian inner ear[1,63]. Many questions remain as to the nature of the direct molecular interactions that lead to such organized ECM patterning. A more systematic analysis of apical ECM components and localization could reveal such mechanisms of patterning and nanoscale organization.

## Methods

### Genetics

*C. elegans* maintenance followed standard procedures[23]. Phenotypic analyses were performed at 20 °C with the exception of temperature-sensitive allele analysis. All available mutant alleles were confirmed by PCR or sequencing. Strains used in this study are listed in Supplementary Data 1.

## Molecular identities of *bli-1*, *bli-2*, and *bli-6*

*bli-1* and *bli-2* map to chromosome II. *bli-1* was localized to an interval defined by the left end of *mnDf90* and the right ends of *mnDf57* and *mnDf58*, a region spanning three cosmid clones. *bli-1* is in a cluster of 4 cuticle collagen genes (from 5′: *col-79, col-39, col-80,* and *bli-1*) within an intron of the *dph-2* gene (transcribed in opposite direction). This genomic organization is also found in *C. briggsae* and *C. brenneri* (Wormbase WS280). *bli-2* is transcribed from the opposite strand of the 4th intron of *zyg-1*; this genomic location is preserved in *C. briggsae* and partially in *C. brenneri*. The molecular identity of *bli-6* was determined by whole genome sequencing of *bli-6(sc16)*.

## CRISPR/Cas9 gene editing

Single-guide RNA Cas9 vectors were cloned using the vector Addgene #47549 and the Quikchange mutagenesis protocol as described[64]. Deletion mutants of *bli-1*, *bli-2*, and *bli-6* were generated by injection of one or two sgRNA plasmids into N2 worms. The *ju1395* deletion also deletes part of exon 5 of *dph-2*, which encodes a diphthamide biosynthetic protein.

For Cas9-mediated knock-in of mNeonGreen (mNG) or other XFPs we followed established protocols[64,65]. See Supplementary Data 1 for primers, plasmids, and flanking sequences for knock-in sites. BLI-2 and BLI-6 were tagged at their C-termini; BLI-1 was tagged either immediately after the subtilisin cleavage site (*ju1789*) or immediately after the Gly-X-Y repeats (*ju1791*). A third *bli-1* knock-in *ju1676* inserted GFP after L737 and caused enhancement of the Bli phenotype of *bli-6(0)*, suggesting a cryptic loss of function; its localization was not examined in detail. The *juEx8105* array contains wrmScarlet[66] inserted at the same site as *ju1789*. Punctate localization of a BLI-1::GFP transgene with GFP inserted after C364 (*cgEx198*) has been reported[31,33] and is consistent with our BLI-1::mNG knock-ins; results here focus on the localization of the N-terminal BLI-1::mNG knock-in *ju1789*, which does not appear to cause cryptic loss of function. The internal knock-in BLI-2::GFP(*ju1678*) caused small blisters at the nose that were exacerbated in a *bli-6(0)* background; *ju1678* localization was not examined further. The BLI-6::mNG knock-in was generated as *ju1914* in wild-type background and as *ju1938* in the *bli-6(mn4)* background. *syb* knock-ins or deletions were purchased from SunyBiotech (Fuzhou, China). The *bli-1* cleavage site mutation *ju1972* and the deletion *ju1980* were induced in the *ju1789* background by the melting method[65]. CRISPR engineered mutations were outcrossed twice to N2 prior to analysis. For quantitation of phenotypic penetrance in Tables 1–3 and Table S1, 50 animals per genotype were scored at 24 h post L4 stage.

## Confocal and widefield light microscopy

Fluorescence imaging was performed on a Zeiss LSM800 or LSM900 confocal using either a Planapo 100x/NA 1.4 OIL DIC M27 objective (pixel xy size 0.0425 μm) or a Planapo 63x/NA 1.4 objective (pixel xy size 0.0675 μm), with Zeiss immersion oil 1.518 (Immersol 518F). Using 100 nm multicolor beads (Zeiss 000000-2114-035) and a Planapo NA 1.4 100x objective on the LSM800, the FWHM at 488 nm was measured as 240 nm in xy and 630 nm in z; at 560 nm the FWHM was 270 nm (xy) and 660 nm (z). For conventional confocal z-stacks, z slices were at 0.29 μm intervals such that the entire cuticle is imaged in 6 z planes ($6 \times 0.29$ μm = 1.7 μm). Pinhole sizes were 0.45–0.72 Airy units, laser power 5–10%, and detector gain set at 500–700 depending on the sample brightness. No background or flatfield correction was applied. Images in Supplementary Fig. 2 used Airyscan SR imaging with a Zeiss Planapo 100x objective NA 1.46, with measured FWHM at 488 nm of 190 nm (xy) and 360 nm (z); at 560 nm the FWHM was 190 nm (xy) and 340 nm (z). Full metadata are in each source image file. Worms were immobilized using 5 mM levamisole in M9. L4 cuticle movies were taken under levamisole anesthetic, typically as z-stacks every 2.5 min, and a single focal plane selected from each z-stack. For precise staging of L4 animals we hatched eggs onto plates lacking food leading to L1

arrest, then re-started development by adding bacterial food. Animals were also staged by L4 vulval morphology[39]. To measure brightness of individual puncta over a movie (Fig. 4f), circular ROIs of 10 μm diameter were selected manually in each movie frame by reference to adjacent landmarks, and average brightness measured using Fiji. for quantitation of colocalization we used the Colocalization Workspace in Zen Blue (Zeiss). Colocalization was analyzed for entire images and for smaller ROIs; data for entire images is presented in Fig. S2e. No zoom or scaling was used. Parameters reported are the Pearson's Correlation Coefficient (R), and weighted Green-Red and Red-Green colocalization coefficients.

P*nlp-29*-GFP(*frIs7*) expression[67] was imaged in animals within 12 h of mid-L4 stage on a Zeiss Axio Imager M2 microscope with GFP-BP filter sets. For quantitation (Fig. 1b), we selected 3 ROIs per animal, of 500 μm² each, along the midbody epidermis. Background was subtracted and fluorescence normalized to wild type.

### HaloTag imaging
A BLI-2::HaloTag C-terminal knock-in *bli-2(syb4687)* (SunyBiotech, Fuzhou, China) was generated using the template plasmid pDD378[68]. HaloTag ligands coupled to Janelia Fluor (JF) 549 or JF646 (Promega)[69] were resuspended in DMSO to a stock concentration of 200 μm and single-use aliquots kept at −20 °C. BLI-2::HaloTag animals were washed off plates in M9 and washed 2–3x in M9 then stained in HaloTag ligand in PBS (final concentration 0.5 μM) at room temperature for 3 h. Animals were washed 3x in M9 and imaged immediately.

### 3D-structured illumination microscopy
Animals were mounted essentially as for standard confocal microscopy (above); in some experiments live animals were immobilized on No. 1.5 gamma-irradiated glass-bottomed dishes (MatTek, Ashland, MA, USA). 3D-SIM was performed on an OMX SR microscope setup (Cytiva, Marlborough, MA, USA) equipped with an Olympus PlanApo N 60X/1.42 oil objective. Using 100 nm multicolor TetraSpeck beads (ThermoFisher) the FWHM at 488 nm was measured to be between 90–100 nm in xy and 310 nm in z. Reconstruction of images was performed using in the OMX software package softWoRx. Reconstruction accuracy was verified using the SIMcheck plugin for ImageJ;[70] all reconstructions were classified as good to very good. The average modulation contrast to noise ratios (MCNRs) in the 3D-SIM reconstructions in Fig. 5 were >8, apart from the DPY-7 reconstructions which scored 4–7.3; an MCNR > 8 is considered good to excellent according to SIMcheck documentation. Visualization of the hollow centers of BLI-containing struts did not require further processing. To smoothen pixelated individual struts (e.g. Figure 5b), images were upscaled with bicubic interpolation (ImageJ). 3D rendering of struts was performed using the open-source software Icy[71].

### Image analysis
For quantitative analysis of strut puncta in adults, we used Fiji to adjust image brightness thresholds and applied the Analyze Particle tool, selecting particles in the range 0.01–2 μm². Size thresholds between 0.01 and 0.02 μm² yielded similar results; size thresholds <0.01 μm² resulted in a large number of small particles being counted as puncta that would not have been scored by manual counts. Particles were measured in ROIs from the lateral and dorsoventral epidermis, with at least 3 ROIs per image.

### Statistical analysis and reproducibility
Data sets were first tested for normality using the Kolmogorov–Smirnov test. Data passing this test were analyzed by parametric tests such as ordinary one-way ANOVA, with a post test determined by the experimental design (e.g. Šidák's multiple comparison test when comparing each experimental group to a control group, Dunn's multiple comparison test when comparing selected groups). Data sets not passing the normality test were analyzed using nonparametric tests i.e. Mann–Whitney test for comparing two data sets or Kruskal–Wallis test for >2 comparisons. All statistical analysis used GraphPad Prism 10; exact P values are stated when reported by the software. All imaging and genetic analyses used stable genetically defined *C. elegans* strains that will be made available for reanalysis upon reasonable request. In general, at least three biological replicates per strain were analyzed independently (i.e. at least 3 animals per genotype imaged). Standard confocal fluorescence images are representative of 5–10 images per condition acquired over at least 3 imaging sessions. Electron microscopy sample sizes are noted below. 3D-SIM data are representative of at least 10 SIM data sets per condition acquired over 5–6 imaging sessions. Dot plots show the mean (red line), and SEM (orange error bar).

### Electron microscopy
Day 1 adult animals were prepared for transmission electron microscopy by high-pressure freezing (HPF) and freeze substitution, essentially as described[72]. The image of L4.8 stage wild-type cuticle in Fig. 4 is from the dorsal cuticle of animal WT_1082-1 grid Q1, imaged at 1.02 nm/px[38]. For quantitation of struts, medial layer, and cortical layer dimensions, we examined 3–5 animals per genotype and ∼100 serial sections per animal, including multiple images of blistered and non-blistered areas per section. Only well-defined struts that extended from cortical and basal layers were scored. As EM sections may not pass through the center of struts, the mean strut width may be a slight underestimate of diameter. Overall, strut width and medial layer thickness in our HPF data are slightly larger than those in historical data[16], possibly reflecting differences in preservation between HPF and glutaraldehyde fixations.

### Biochemistry
All biochemical analysis used mNG knock-in strains and anti-mNG monoclonal antibodies (ChromoTek 32F6). Animals were grown at 20 °C as synchronized populations following bleach treatment and collected at different L4 substages as indicated by vulval morphology. Extracts of soluble cuticle proteins were prepared following previously described methods[16] with minor modifications. Briefly, worms were collected using M9J buffer in a 15 ml Falcon tube and washed 3X with M9J buffer. Worm pellets were resuspended in 5 ml of sonication buffer and incubated on ice for 10 min, sonicated (Misonix XL-2000 Series Ultrasonic Liquid Processor, 10 × 10 s pulse with 10 s break) with 30 μl of 0.1 M PMSF. Following sonication, samples were centrifuged at 6010 g for 2 min at 4 °C and supernatants stored as F1 (intracellular proteins) at −20 °C. Pellets were washed 3X with sonication buffer and the resultant pellets were resuspended in 100 μl sonication buffer, boiled at 95 °C for 2 min with 1 ml of ST buffer and incubated overnight with rotation at RT. Following incubation, samples were centrifuged at 6010 g for 3 min and supernatants were stored as F2 (organelle fraction) at −20 °C. Subsequently, cuticle pellets were washed 3X with 0.5% triton X-100 and boiled at 95 °C for 2 min with 300 μl of ST buffer with 5 % of β-ME and incubated with rotation at RT for 16 h. Finally, the samples were centrifuged at 6010 g for 3 min and supernatants collected and stored as F3 (soluble cuticle proteins).

Fractions were precipitated with methanol/chloroform and resuspended in 100 μl of rehydration buffer from which 30 μl was boiled with 10 μl of 4X Laemmli buffer for 5 min and run on 7.5% precast polyacrylamide gel for 60 min then transferred to 0.45 micron nitrocellulose membrane (Bio-Rad) at 4 °C for 90 min using Bio-Rad WET transfer system. For gradient gels, 20 μl samples were loaded along with 7 μl of 4X NuPAGE LDS sample buffer as described by the manufacturer onto NuPAGE 4 to 12% Protein Gels (Invitrogen: NP0322BOX). Following transfer, membranes were blocked in 5% skim milk for 1 h and washed 3X with TBST 5 min each. After washing the membrane was incubated overnight in 1° Ab at 1:5000 dilution at 4 °C on rocker. Membranes were

washed 3X with TBST and incubated in 2° Ab (Goat anti-Mouse IgG, Sigma 12-349) at 1:100,000 dilution at RT on rocker for 1 h and washed 3X with TBST. Blots were visualized with Supersignal West Femto detection kit (ThermoFisher) using a LI-COR Odyssey imager. At least three biological replicates were performed for each blot.

### Reporting summary

Further information on research design is available in the Nature Portfolio Reporting Summary linked to this article.

## Data availability

Further information and requests for resources should be directed to and will be fulfilled by the lead contact, Andrew Chisholm (adchisholm@ucsd.edu). Source image files have been uploaded to Figshare at https://doi.org/10.6084/m9.figshare.24353035. Source data are provided with this paper.

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

## Acknowledgements

We thank members of the Chisholm and Jin labs for discussion and Jordan Ward for comments on the manuscript. We thank the *Caenorhabditis* Genetics Center (CGC) and Mario de Bono (MRC-LMB, Cambridge, UK) for *bli* mutant stocks. We thank Jonathan Hodgkin and Delia O'Rourke (University of Oxford) for sharing their information on the molecular identity of *bli-6*. We thank David Hall and Ken Nguyen of the Center for *C. elegans* Anatomy for sharing the L4 cuticle EM image and Geert de Vreede (Zeiss) for advice on PSF calibration. We thank Brian Ackley for help locating historical data. We thank Laura Toy for sequencing of mutations and generation of some CRISPR deletion alleles, as well as Jade Diaz and Risa Iwazaki for help with strain construction. S.L.Z. was supported by UCSD TRELS; we thank Dan Starr (UC Davis) for hosting S.L.Z. during part of this work. The Center for *C. elegans* Anatomy is funded by NIH R24 OD010943 and the Caenorhabditis Genetics Center is funded by NIH P40 OD10440. Funding was provided

by HHMI gift funds to Y.J., NIH R35 GM142433 to A.M.E., and NIH R01 GM054657 and R35 GM134970 to A.D.C.

## Author contributions

J.G.A., M.P., S.L.Z., and J.R.C. designed and performed C. elegans molecular genetics and analyzed data. E.M.J. performed HaloTag staining. M.P. performed biochemical analyses and confocal microscopy. A.M.E. performed super-resolution microscopy. A.G. performed electron microscopy. Y.J., J.M.K., A.M.E., and A.D.C. secured funding. J.G.A., M.P., and A.D.C. wrote the paper with input from other authors.

## Competing interests

The authors declare no competing interests.
