## [Peer Review File · Nature Communications]

Nanoscale patterning of collagens in *C. elegans* apical extracellular matrixReviewer #1 (Remarks to the Author):

In the submitted work, "Nanoscale patterning of collagens in *C. elegans* apical extracellular matrix" Adams and colleagues, conduct a thorough analysis of three collagen encoding blistering (*bli*) genes, *bli-1*, *bli-2*, and *bli-6*. They undertake a comprehensive analysis of mutant phenotypes (plate level and transmission electron micrographs of the cuticle), tag each gene endogenously with mNeonGreen fluorophore, and characterize their formation into struts in the *C. elegans* cuticle. Using 3D-SIM, they reveal a cylindrical architecture of the struts. They go on to perform genetic interaction studies, revealing similar roles for *BLI-1* and *BLI-2* in struts, but a distinct and perhaps modulatory role for *BLI-6*, and they examine genetic interactions with other cuticle regulating genes, revealing a role for cuticle furrows in patterning struts.

We understand very little about how complex patterns are generated in apical ECM. The blistered phenotype was first identified by Sydney Brenner and this is the most comprehensive study of these genes and their role in cuticle to date. Strengths of the paper include the thorough characterization of *bli-1*, -2 and -6 mutants, characterization of strut formation by endogenously tagging *BLI-1*, -2 and -6, the use of 3D-SIM to reveal struts appear to form into tubes, that struts extend into the fibrous layer of the cuticle and pattern holes here, and the use of genetics to look at genetic interactions between *bli-1*, -2 and -6 as well as other cuticle genes, revealing that furrow in cuticle play a key role in strut organization and patterning. The paper is also clear and well written.

There are, however, several significant weaknesses of the work.

(1) One is that it is unclear general principle(s) can be taken from this work that can help inform our understanding of apical ECMs in other contexts. For example, do these collagens function similarly in other apical ECMs in the worm, such as the gut, vulval, or uterine matrices? Are strut-like structures seen in vertebrate apical ECMs? It would be helpful to have this elaborated in the discussion. Currently it is not, and thus it is unclear if this work will be of broad interest or impact to the field.

(2) Second, there is a significant lack of quantification (number of animals examined, frequency of phenotypes observed, and statistical analysis of those observations) and thus rigor throughout the work.

Examples include:

- > No quantification of how many animals examined for each allele in Tables 1-4.
- > Numbers/quantification for *pnpl-29*-GFP expression in *Bli* mutants for figure 1B are unclear.
- > Where is quantification (#animals, cases examined) for this statement (line 158 results):

Struts are pillar-like structures oriented perpendicular to the plane of the cuticle and appear to connect the cortical and basal layers. The medial layer is typically 150-200 nm thick in young adults; struts are about 250-300 nm in diameter and spaced at least as far apart along the anteroposterior and circumferential axes.

- > Where is quantification for these statements (line 163 results):

All *bli* mutants examined displayed aberrant or missing struts in TEM. Cortical and basal cuticle layers were typically recognizable but sometimes aberrant, suggesting the *Bli* phenotype reflects an initially specific disruption of strut or medial layer formation that may lead to secondary disruption of other layers.

In *bli-1(lf)*, *bli-2(lf)*, and *bli-6(gf)* mutants the medial layer lacked struts and instead contained loose electron dense granular or occasionally fibrous material (Figure 2B-D), consistent with a study of *bli-1(m361)* (de Melo et al., 2003).

> For Figure 2, expansion of the medial layer is not mentioned in the results text, but is in the legend. Is it possible to quantify this? The text also mentions that the cortical layer is mostly normal, but it looks thinner in the pictures (visibly thinner in B, electron dense in C, not in field of view in D). Could the authors resolve these discrepancies?

> Where is quantification for these statements (line 275 results):

Like DPY-13::mNG the most apical layer of COL-19::mNG shows regularly spaced holes similar in size and spacing to struts (Figure S2 E). The regularly spaced holes in the wild-type COL-19::mNG fibrous layer were absent in bli-1 and in bli-6(gf) animals, although other aspects of annular or fibrous layer organization appeared normal.

In bli-6(0) mutants, COL-19::mNG holes were altered in distribution such that holes in the center of each annulus were enlarged and lateral furrow-flanking gaps were smaller, correlating with the altered distribution of BLI-1::mNG puncta in bli-6(0) mutants (Figure S2 H).

> For Figure 3, do the authors have a co-localization metric? For example, in Figure 3E, it would be helpful to at least state that this was seen in x/x animals. For Figure 3F a co-localization metric would be important. I have trouble distinguishing what fraction of BLI-6 puncta co-localize with BLI-1.

> Where is quantification for these statements (line 305 results):

Puncta in the central annular row formed at a more variable rate; some large distinct puncta could be observed at 41 h after release from L1 arrest. Individual puncta appeared to form locally from diffuse BLI-1::mNG, and once formed did not further move, fuse, or separate (e.g. Figure 4D).

Each individual punctum formed over a period of 5-10 minutes and subsequently became brighter over ~ 1 h (Figure 4D), but due to asynchrony the overall pattern takes > 1 h to develop under our imaging conditions. Individual puncta were variable in shape during their initial formation but subsequently became consistently circular in cross-section. In rare cases large puncta appeared to form by fusion of adjacent puncta (arrowhead, Figure 4E).

The BLI-2::mNG signal was weaker than that of BLI-1::mNG, but generally displayed similar dynamics in the L4 stage. BLI-2::mNG initially formed patchy furrow flanking bands (~40 h post L1 arrest) which over a period of 1-2 h condensed into puncta resembling those in the mature adult cuticle (Figure 4B; Video 2).

BLI-6::mNG showed similar dynamics, although it was more concentrated in the cuticle of the lateral epidermis in early L4 stages (Figure 4C).

> Where is quantification for this statement (Supplemental Figure legend 3 (line 53):

Nascent struts (red arrowheads) form adjacent to furrow bases or in the center of annuli and are ~100 nm diameter at this stage; a medial layer is visible, but not a basal layer. Scale, 500 nm.

> Where is quantification for these statements (Results line 351):

Using 3D-SIM (see Methods), we found that BLI-1, BLI-2, and BLI-6 mNG knock-ins localized to cylindrical or conical structures, appearing as hollow rings in cross-section (Figure 5A,B). Each cylinder had one or more non-fluorescent centers of 100-150 nm in diameter surrounded by a ring of fluorescence. Some struts displayed double rings or 'figure 8' appearances in a z-section; others appeared to be formed of partial cylinders. These cylindrical structures extended ~500 nm orthogonal to the cuticle plane, consistent with the vertical dimensions of struts in EM and confocal microscopy.

> Where is quantification for these statements (Results line 385):

bli-1(0) mutants typically progressed from mild Bli at 3 h post-molt to severe Bli at 9 h post-molt.

In contrast, blistering in the bli-1(0); bli-6(ju1681) double mutant was intermediate by 3 h post-molt and progressed to severe by 5 h post-molt; blisters were also larger in double mutants at 24 and 48 h post-molt (Figure 6A). bli-6(ju1681) similarly enhanced bli-2(0) in that bli-2(0) single mutants typically developed a single large head blister and several smaller blisters near the tail, whereas double mutants displayed smaller head blisters and more mid-body blisters; double mutants were often deceased by 48 h after the L4/adult molt, unlike bli-2(0) single mutants (not shown).

We next examined whether BLI proteins require one another for incorporation into the cuticle. In bli-2(0) mutants, very low levels of BLI-1::mNG were observed in the cuticle (Figure 6B); scattered puncta of varying sizes could be seen in unblistered regions whereas in blisters, BLI-1::mNG localized to diffuse bands approximately the size of annuli on the external (cortical) side of the blister and to rare scattered puncta that were usually brighter than wild type BLI-1::mNG puncta. These observations suggest BLI-2 is required for regular BLI-1 punctate localization and is important in promoting BLI-1 secretion into the cuticle. Conversely, in bli-1(0) mutants BLI-2::mNG showed minimal punctate localization, but instead localized to faint circumferential bands on either side of furrows (Figure 6E). BLI-2 localization to lateral alae appeared normal in bli-1(0) mutants. These data suggest BLI-1 is required for BLI-2 localization and organization into struts; in the absence of BLI-1, a small amount of BLI-2 is present in the cuticle but mostly remains in diffuse furrow-flanking bands. In bli-6(0) mutants, BLI-1::mNG and BLI-2::mNG localized to rows of puncta, but the distribution was altered such that the central row puncta were brighter relative to the furrow-flanking rows (Figure 6C,F). As BLI-6 itself appears enriched in the central row of puncta (see below), BLI-6 may act locally to negatively regulate BLI-1 localization. Notably, in bli-6(mn4) gain of function mutants, BLI-1::mNG and BLI-2::mNG localized to disorganized puncta in dorsoventral (muscle adjacent) regions, outside blistered areas (Figure 6D,G)

Unlike BLI-1::mNG and BLI-2::mNG, BLI-6::mNG was present in both cortical and basal layers in blisters (Figure 6H-J). Few BLI-6::mNG puncta were visible in bli-1(0) mutants and the remaining BLI-6::mNG localized to cortical furrow-like bands over the blistered areas although some variable aggregates were also observed inside the blister (Figure 6I, J).

(3) Third, there are several instances where results are described with no supporting data. Please provide all data that accompany statements within results.

Examples include:

> Lines 186-189: The text mentions diffuse signal of BLI-1 in the mid-L4, please show this data. Please also show the localization of the N-terminal construct, which is stated to look like the C-terminal construct.

> Lines 210-219, 220-221: Where are the supporting images for these statements:

Rows of BLI-1::mNG puncta were observed over almost all the lateral and dorsoventral cuticle, with mNG brightness varying depending on the region. Puncta in the head and the dorsal tail cuticle (overlying epidermal cells hyp5 and hyp10) were brighter than in other regions. BLI-1::mNG puncta overlying the edges of body muscle quadrants were also brighter than those in non-muscle-adjacent epidermis, suggesting that within each epidermal cell, deposition of BLI-1 might be modulated by local mechanical forces or signals from muscle.

other cuticle regions containing little or no BLI-1::mNG include the anterior tip of the head (external to hyp4), the hermaphrodite vulva, and the excretory pore.

> Lines 265-266: Where are the pictures showing BLI-1 in gaps of DPY-13 signal?

(4) Fourth, it would be helpful to make the manuscript more accessible to readers not familiar with *C. elegans*. For example, the L4 cuticle, nascent adult cuticle, and young adult cuticles are described, but stages examined are not clearly labelled in figures. A schematic of cuticle development, and definition of key terms such as alae and cuticle furrows early in the manuscript will be very helpful.

Minor comments:

> Figure 3, Figure 5—please avoid using red, as many readers are red-green color blind. Please use a color-blind friendly pairing, such as cyan and magenta.

> Figure 1C. Authors have C terminus on left and N terminus on right (?) of protein. This goes against standard convention of how proteins are drawn out—N terminus is on left and C terminus is on right

Reviewer #2 (Remarks to the Author):

Adams et al. present a study on the patterns of collagen localization in the cuticle of *C. elegans*. Using a variety of approaches, they suggest the presence of nanoscale patterning. I was tasked to provide a technical review for this manuscript and will focus my comments on the author's use and analysis of microscopy approaches.

This work utilized confocal, widefield, and 3DSIM microscopy combined with several fluorescence labeling strategies. In general, standard technical information and validations for complex microscopy experiments are lacking in this work. For example, the author's "nanoscale patterning" claim relies on the colocalization of fluorescence labels, but microscopy information, colocalization quantification, and controls are missing.

For standard microscopy information, the lateral and axial sampling is missing for all methods, the confocal microscope and widefield microscope's optical configuration (e.g., the numerical aperture), the exposure times are missing, and any background or flatfield corrections applied in the acquisition software are not reported. Without this information, it is not possible to evaluate the colocalization claims.

For colocalization analyses, two valuable guides for evaluating colocalization are doi: 10.1152/ajpcell.00462.2010 and doi: 10.1038/s41596-020-0313-9. Quantifying the colocalization, through multiple approaches, using the 3D stacks would help strengthen the case, which can be performed using Fiji.

For the 3DSIM data, the authors do not present a multicolor calibration of the instrument using the specific refractive index matching oils and sample refractive index used in the experiments. The authors should describe the exact alignment and quality check protocols to ensure colocalization analysis is possible using the data. The modulation contrast to noise ratio (MCNR) and the estimated resolutions in XYZ are two useful parameters to report. Resolutions should be reported using diffraction-limited emitters during calibration and estimated from the data using Fourier ring correlation or Image decorrelation analysis. Both are available as Fiji plugins.

In addition to single-color bead samples for calibration, 3DSIM data for multicolor beads should be present to ensure that the recovered super-resolution images are of sufficient resolution and quality to support the author's colocalization claims.

These checks are essential, as the patterns in Figure 5D resemble "hammerstroke" noise, commonly present in noise-limited SIM reconstructions. In addition, the 3D renderings in Figure 5 are a bit puzzling. Can the authors please explain the lack of detail in the axial dimension for Figures 5B and 5E?

Beyond the microscopic methods, I have two additional points I would ask the authors to clarify:

-Why are Figures 6 and 7 presented with inverted lookup tables, while the rest of the manuscript is present with standard lookup tables?

-For the image analysis, did the authors explore how varying the threshold around their chosen

threshold altered their puncta counting results? The value should be robust to perturbations for the proposed analysis to be valid.

RESPONSE TO REVIEWERS

Reviewer comments are in italics, quoted text is in romans, and our responses are in bold face; we have numbered the reviewer comments for ease of reference.

Reviewer #1 (Remarks to the Author):

*In the submitted work, “Nanoscale patterning of collagens in *C. elegans* apical extracellular matrix” Adams and colleagues, conduct a thorough analysis of three collagen encoding blistering (*bli*) genes, *bli-1*, *bli-2*, and *bli-6*. They undertake a comprehensive analysis of mutant phenotypes (plate level and transmission electron micrographs of the cuticle), tag each gene endogenously with mNeonGreen fluorophore, and characterize their formation into struts in the *C. elegans* cuticle. Using 3D-SIM, they reveal a cylindrical architecture of the struts. They go on to perform genetic interaction studies, revealing similar roles for *BLI-1* and *BLI-2* in struts, but a distinct and perhaps modulatory role for *BLI-6*, and they examine genetic interactions with other cuticle regulating genes, revealing a role for cuticle furrows in patterning struts.*

*We understand very little about how complex patterns are generated in apical ECM. The blistered phenotype was first identified by Sydney Brenner and this is the most comprehensive study of these genes and their role in cuticle to date. Strengths of the paper include the thorough characterization of *bli-1*, -2 and -6 mutants, characterization of strut formation by endogenously tagging *BLI-1*, -2 and -6, the use of 3D-SIM to reveal struts appear to form into tubes, that struts extend into the fibrous layer of the cuticle and pattern holes here, and the use of genetics to look at genetic interactions between *bli-1*, -2 and -6 as well as other cuticle genes, revealing that furrow in cuticle play a key role in strut organization and patterning. The paper is also clear and well written.*

We thank the reviewer for the positive comments.

There are, however, several significant weaknesses of the work.

(1) One is that it is unclear general principle(s) can be taken from this work that can help inform our understanding of apical ECMs in other contexts. For example, do these collagens function similarly in other apical ECMs in the worm, such as the gut, vulval, or uterine matrices? Are strut-like structures seen in vertebrate apical ECMs? It would be helpful to have this elaborated in the discussion. Currently it is not, and thus it is unclear if this work will be of broad interest or impact to the field.

We appreciate the reviewer’s comment and have expanded our discussion of the general principles that can be derived from our analysis of the BLI collagens (lines 791-797 of revised discussion). We clarify (lines 199-201) that expression of the 3 BLI collagens as judged by knock-in markers is restricted to late L4 and adult epidermis, consistent with their mutant phenotypes. We have added additional images (new Figure S2A-D) of BLI-2 and BLI-6 localization to cuticle of interfacial tubes such as the vulva and sensory pores, which unlike most adult cuticle are unlayered and lack struts (WormAtlas, <https://www.wormatlas.org/hermaphrodite/cuticle/Cutframeset.html>). As BLI-2 and BLI-6 localize to multiple cuticle types and substructures other than struts, they may have additional roles that we have not explored in the present work. Other aECMs such as those of the *C. elegans* pharyngeal and buccal capsule display nanoscale structural

specializations such as sieve extensions, grinder teeth, and circumferential ribs; these are most likely made of chitin and/or pharynx-specific ECM proteins (Kamal et al., 2022). The mechanisms by which these nanoscale structures are patterned are also not known. The aECMs of the *C. elegans* gut and uterus appear to be non-collagenous glycoalyx type layers (WormAtlas) and remain less well characterized; some collagens may be expressed in the intestine but their ECM localization has not been determined. Although struts have not been observed in aECMs produced by these internal epithelia, we believe analysis of struts can exemplify general principles of how ECM can be patterned on the nanoscale, whether by collagens or other ECM molecules.

The terminology 'apical ECM' is not widely used in the vertebrate ECM literature, however there are several examples of pillar or tube-like ECM structures on the nanoscale which we have now mentioned in the discussion. The most well-known is the marginal pillars/marginal net of the vertebrate inner ear aECM (tectorial membrane). Vertebrate heart muscle cells are connected by pillar-like endomysial struts, thought to be collagenous. We hope reviewers will agree our work will have impact by stimulating more studies of vertebrate ECM architecture.

As shown in tracked text, we have clarified in lines 763-766 of revised discussion that strut-like structures (defined as connecting cuticle layers) are seen in many nematode cuticles; struts are relatively simple examples of these ECM specializations. Our work is the first molecular genetic analysis of such structures in any nematode. As nematodes are widespread parasites of humans, animals, and plants, as well as being major components of the biome, better understanding of nematode biology can have significance to human health.

(2) Second, there is a significant lack of quantification (number of animals examined, frequency of phenotypes observed, and statistical analysis of those observations) and thus rigor throughout the work.

Examples include:

Comment 2.1 > No quantification of how many animals examined for each allele in Tables 1-4.

We have added sample size information to the Tables (n >= 50 animals per genotype) and have indicated % penetrance where phenotypes are not fully penetrant.

*Comment 2.2 > Numbers/quantification for *pnpl-29-GFP* expression in *Bli* mutants for figure 1B are unclear.*

We have added a graph to Figure 1B showing quantitation of fluorescence intensity. Both *bli* mutants show highly significant *Pnlp-29-GFP* upregulation compared to wild type.

Comment 2.3 > Where is quantification (#animals, cases examined) for this statement (line 158 results):

"Struts are pillar-like structures oriented perpendicular to the plane of the cuticle and appear to connect the cortical and basal layers. The medial layer is typically 150-200 nm thick in young

adults; struts are about 250-300 nm in diameter and spaced at least as far apart along the anteroposterior and circumferential axes.”

We have quantitated dimensions of struts, medial layer, and cortical layer in our HPF EM datasets, now presented in Figure 2E. We find a mean strut width of 194 ± 7 nm (SEM, n = 61), range 109-392 nm, and mean medial layer thickness of $221 \text{ nm} \pm 11 \text{ nm}$ (n = 70). We have added a note to the Methods (lines 929-939) that this may underestimate strut width given EM sections often do not pass through the center of struts. We scored only struts that connected cortical and basal layers. In comparison, published images of longitudinally sectioned animals such as those from Cox et al. 1981a (J. Cell Biol) using Glutaraldehyde fix and OsO₄ stain show a slightly lower mean strut width of 172 nm (n = 8 struts, our quantitation of the Cox et al EM is shown in Fig 2E). Images of struts in the WormImage EM archive (<https://www.wormimage.org>) are consistent with our quantitation, however as WormImage datasets often lack scale information or details on animal age / growth condition we have not pursued their detailed quantitation. Overall the medial layer is thicker in our HPF data sets than in glutaraldehyde-fixed animals, possibly reflecting differential preservation of the fluid-filled medial layer.

We have also clarified in Methods that our ultrastructural analysis is based on serial sections of 3-5 animals per genotype, ~100 sections per animal, and multiple images per section including blistered and non-blistered areas.

Comment 2.4 > Where is quantification for these statements (line 163 results):

“All bli mutants examined displayed aberrant or missing struts in TEM. Cortical and basal cuticle layers were typically recognizable but sometimes aberrant, suggesting the Bli phenotype reflects an initially specific disruption of strut or medial layer formation that may lead to secondary disruption of other layers.”

C. elegans electron microscopy data sets often involve a small number of animals, so qualitative summaries are typically accepted (cf. e.g. Katz et al 2022, or Aggad et al 2023) as a guide for quantitative investigations by other approaches. We have clarified (lines 173-175) that we did not observe any struts in >500 sections of bli-1(e944) or bli-2(e768) nor in >300 sections of bli-6(n776). We have quantified cortical layer thickness (Figure 2E) as detailed in response to comment 2.5A. In our HPF sections the fibrous layer appears uneven such that frayed fibrous material extends into the medial layer. We have therefore not quantitated fibrous layer thickness. However bli mutants display two distinct fibrous layers with a ‘herringbone’ appearance similar to the wild type (Figure 2).

“In bli-1(lf), bli-2(lf), and bli-6(gf) mutants the medial layer lacked struts and instead contained loose electron dense granular or occasionally fibrous material (Figure 2B-D), consistent with a study of bli-1(m361) (de Melo et al., 2003).”

The images shown are representative of 4/5 bli-2 animals which display granular material in the expanded medial layer. 1/5 bli-2(e768) animals displayed fibrous/filamentous material; as this was a low penetrance phenotype we have not pursued it further.

Comment 2.5A > For Figure 2, expansion of the medial layer is not mentioned in the results text, but is in the legend. Is it possible to quantify this?

We have quantitated the medial layer in the EM and have added this data to Figure 2E. Note that in severely blistered *bli-1* mutants the medial layer is frequently expanded to a degree (>50 μm) that EM images do not capture the entire layer, thus EM-based quantitation underestimates the degree of expansion. We have clarified (lines 178-180) that in *Bli* mutant EM the medial layer is variable in thickness: in blistered regions it is greatly expanded but in some non-blistered regions it is sometimes thinner (Figure 2E). This results in an apparently bimodal distribution of medial layer thicknesses in *bli-1* and *bli-2* mutants (Figure 2E).

> Comment 2.5B: The text also mentions that the cortical layer is mostly normal, but it looks thinner in the pictures (visibly thinner in B, electron dense in C, not in field of view in D). Could the authors resolve these discrepancies?

The reviewer is correct that the cortical layer is thinner in *bli-1* and *bli-2* mutants and we have added quantitation of the EM to Figure 2 to clarify this point. The cortical layer otherwise appears normal in morphology. For EM of *bli-6(n776)* we measured the cortical layer in blistered and non-blistered regions and find that the cortical layer is thicker in non-blistered regions compared to blisters (new data in Figure 2E). Thinning of the cortical layer may be a secondary consequence of inflation of the blister and consequent expansion of the cortical layer, although it remains possible that struts somehow contribute directly cortical layer formation. The *BLI* collagens do not localize to the cortical layer, nor do the other collagens described here. We plan to address the architecture of the cortical layer in future investigations.

Comment 2.6 > Where is quantification for these statements (line 275 results):

“Like *DPY-13::mNG* the most apical layer of *COL-19::mNG* shows regularly spaced holes similar in size and spacing to struts (Figure S2 E). The regularly spaced holes in the wild-type *COL-19::mNG* fibrous layer were absent in *bli-1* and in *bli-6(gf)* animals, although other aspects of annular or fibrous layer organization appeared normal.

“In *bli-6(0)* mutants, *COL-19::mNG* holes were altered in distribution such that holes in the center of each annulus were enlarged and lateral furrow-flanking gaps were smaller, correlating with the altered distribution of *BLI-1::mNG* puncta in *bli-6(0)* mutants (Figure S2 H).”

We have quantified the ‘holes’ in the surface *COL-19::mNG* layer and present the data in the revised Fig S3 I,J. In circumferential line scans along the centers of annuli these appear as periodic troughs in the intensity of the *COL-19::mNG* signal. Spacing of these troughs in line scans along circumferential axis in the center of annuli (0.79 μm) is consistent with strut spacing of ~0.77 μm in the circumferential axis. These troughs are not seen in line scans of *bli-1* null mutants or *bli-6* GOF mutants. Note that such ‘holes’ are not as precisely defined as struts due to the fibrous nature of *COL-19::mNG* distribution. Manual quantitation of *COL-19::mNG* hole diameter indicates they are slightly larger than struts (~400 nm) in the wild type, and significantly larger in *bli-6* null mutants.

Comment 2.7 > For Figure 3, do the authors have a co-localization metric? For example, in Figure 3E, it would be helpful to at least state that this was seen in x/x animals. For Figure 3F a

co-localization metric would be important. I have trouble distinguishing what fraction of BLI-6 puncta co-localize with BLI-1.

We have added quantitative colocalization metrics to a new Supplementary Figure 2. We show the Pearson Correlation Coefficient (R) for experiments in Figures 3 and 5, as well as the individual channel (Red-Green and Green-Red) weighted colocalization coefficients. As positive controls BLI-1::wrmSc was highly correlated (R = 0.88) with BLI-1::mNG in heterozygous animals. BLI-1::mNG and BLI-2::HT correlation was lower (R = 0.40). BLI-1 and BLI-6 did not significantly colocalize in analysis of large ROIs likely because BLI-6::mNG displays diffuse signal and additionally localizes to furrows and alae. BLI-1::mSc and BLI-6::mNG displayed variable but positive colocalization as judged by the channel colocalization coefficients. BLI-2::HT displayed low colocalization with DPY-9 or DPY-13 mNG reflecting overlapping localization with furrows and annuli.

We also include intensity line scans (new Figure S2F) as representative examples supporting that BLI-1::mSc and BLI-2::mNG peaks coincide whereas BLI-1::mSc peaks coincide with DPY-13::mNG troughs.

Comment 2.8 > Where is quantification for these statements (line 305 results):

“Puncta in the central annular row formed at a more variable rate; some large distinct puncta could be observed at 41 h after release from L1 arrest. Individual puncta appeared to form locally from diffuse BLI-1::mNG, and once formed did not further move, fuse, or separate (e.g. Figure 4D).

“Each individual punctum formed over a period of 5-10 minutes and subsequently became brighter over ~ 1 h (Figure 4D), but due to asynchrony the overall pattern takes > 1 h to develop under our imaging conditions. Individual puncta were variable in shape during their initial formation but subsequently became consistently circular in cross-section. In rare cases large puncta appeared to form by fusion of adjacent puncta (arrowhead, Figure 4E).

“The BLI-2::mNG signal was weaker than that of BLI-1::mNG, but generally displayed similar dynamics in the L4 stage. BLI-2::mNG initially formed patchy furrow flanking bands (~40 h post L1 arrest) which over a period of 1-2 h condensed into puncta resembling those in the mature adult cuticle (Figure 4B; Video 2).

“BLI-6::mNG showed similar dynamics, although it was more concentrated in the cuticle of the lateral epidermis in early L4 stages (Figure 4C).”

We have quantitated brightness of ROIs corresponding to individual BLI-1::mNG puncta formation from Video 1 data (Figure 4F). The quantitative analysis indicates that initial condensation into puncta takes 5-10 minutes and corresponds to a rapid increase in mean fluorescence, which thereafter increases slowly before plateauing. Due to sample movement in these Video sequences, ROIs were selected manually at each frame. Additionally, due to the increased sample movement and lower signal to noise in the BLI-2::mNG video it was not possible to consistently quantitate single BLI-2::mNG puncta over time.

We have added L4 developmental substage timing information to the Figure 4 legend, in addition to the information on chronological timing.

Comment 2.9 > Where is quantification for this statement (Supplemental Figure legend 3 (line 53): Nascent struts (red arrowheads) form adjacent to furrow bases or in the center of annuli and are ~100 nm diameter at this stage; a medial layer is visible, but not a basal layer. Scale, 500 nm.

We have quantitated strut width in the L4 stage EM and have added this to the revised Supplementary Figure 4 legend; the mean is 107 ± 5 nm ($n = 37$). In the L4.8/9 stage EM strut electron densities are less well-defined and quantitation is approximate. Some struts are not clearly separated from larger regions of electron density between the base of furrows and fibrous layer. These may represent initial stages in strut generation; we only quantitated struts if they were separated from these regions. Overall there is limited ultrastructural data on L4 struts as analysis of their biogenesis relative to furrows requires longitudinal sectioning for EM, which is rarely performed. The images quantitated are from a single available animal generously provided by D. Hall (mentioned in Acknowledgements).

Comment 2.10 > Where is quantification for these statements (Results line 351):

“Using 3D-SIM (see Methods), we found that BLI-1, BLI-2, and BLI-6 mNG knock-ins localized to cylindrical or conical structures, appearing as hollow rings in cross-section (Figure 5A,B). Each cylinder had one or more non-fluorescent centers of 100-150 nm in diameter surrounded by a ring of fluorescence. Some struts displayed double rings or ‘figure 8’ appearances in a z-section; others appeared to be formed of partial cylinders. These cylindrical structures extended ~500 nm orthogonal to the cuticle plane, consistent with the vertical dimensions of struts in EM and confocal microscopy.”

We have added quantitation of BLI ring diameters to Figure 5B, as well as a representative line scan. The mean diameter of struts (defined as peak-to-peak fluorescence in line scans) exhibiting non-fluorescent centers was $161 \text{ nm} \pm 29 \text{ SD}$ ($n = 110$) for BLI-1::mNG, and $150 \text{ nm} \pm 22 \text{ SD}$ ($n = 86$) for BLI-2::mNG, which falls within the estimated resolution of 3D-SIM reconstructions of $< 110 \text{ nm XY}$. BLI-1::mNG ring diameters are slightly wider than those of BLI-2::mNG.

Comment 2.11 > Where is quantification for these statements (Results line 385):

“bli-1(0) mutants typically progressed from mild Bli at 3 h post-molt to severe Bli at 9 h post-molt. In contrast, blistering in the bli-1(0); bli-6(ju1681) double mutant was intermediate by 3 h post-molt and progressed to severe by 5 h post-molt; blisters were also larger in double mutants at 24 and 48 h post-molt (Figure 6A). bli-6(ju1681) similarly enhanced bli-2(0) in that bli-2(0) single mutants typically developed a single large head blister and several smaller blisters near the tail, whereas double mutants displayed smaller head blisters and more mid-body blisters; double mutants were often deceased by 48 h after the L4/adult molt, unlike bli-2(0) single mutants (not shown).”

We have added a statement to Figure 6 legend (lines 1237-8) that these are based on longitudinal observations of 10 animals observed longitudinally per genotype.

“We next examined whether BLI proteins require one another for incorporation into the cuticle. In *bli-2(0)* mutants, very low levels of BLI-1::mNG were observed in the cuticle (Figure 6B); scattered puncta of varying sizes could be seen in unblistered regions whereas in blisters, BLI-1::mNG localized to diffuse bands approximately the size of annuli on the external (cortical) side of the blister and to rare scattered puncta that were usually brighter than wild type BLI-1::mNG puncta. These observations suggest BLI-2 is required for regular BLI-1 punctate localization and is important in promoting BLI-1 secretion into the cuticle. Conversely, in *bli-1(0)* mutants BLI-2::mNG showed minimal punctate localization, but instead localized to faint circumferential bands on either side of furrows (Figure 6E). BLI-2 localization to lateral alae appeared normal in *bli-1(0)* mutants. These data suggest BLI-1 is required for BLI-2 localization and organization into struts; in the absence of BLI-1, a small amount of BLI-2 is present in the cuticle but mostly remains in diffuse furrow-flanking bands. In *bli-6(0)* mutants, BLI-1::mNG and BLI-2::mNG localized to rows of puncta, but the distribution was altered such that the central row puncta were brighter relative to the furrow-flanking rows (Figure 6C,F). As BLI-6 itself appears enriched in the central row of puncta (see below), BLI-6 may act locally to negatively regulate BLI-1 localization. Notably, in *bli-6(mn4)* gain of function mutants, BLI-1::mNG and BLI-2::mNG localized to disorganized puncta in dorsoventral (muscle adjacent) regions, outside blistered areas (Figure 6D,G).”

We have now quantitated BLI puncta size in *bli* mutants (main text and new Figure 6H). The few remaining BLI-1::mNG puncta in *bli-2* mutants are larger than in wild type. Puncta in *bli-6(gf)* and *bli-6(0)* mutants are more variable but on average larger than in wild type.

“Unlike BLI-1::mNG and BLI-2::mNG, BLI-6::mNG was present in both cortical and basal layers in blisters (Figure 6H-J). Few BLI-6::mNG puncta were visible in *bli-1(0)* mutants and the remaining BLI-6::mNG localized to cortical furrow-like bands over the blistered areas although some variable aggregates were also observed inside the blister (Figure 6I, J).”

BLI-6 aggregates in the *bli-1* mutants are not readily quantitated by automated particle analysis (as in Fiji) because they are sparse and surrounded by diffuse signals as well as remaining localization to other structures. Wild type BLI-6::mNG is also not readily quantitated by automated particle counting due to its localization to multiple structures as well as diffuse signal. We believe the abnormal BLI-6 pattern is better conveyed as a qualitative observation (revised lines 472-476).

(3) Third, there are several instances where results are described with no supporting data. Please provide all data that accompany statements within results. Examples include:

Comment 3.1 > Lines 186-189: The text mentions diffuse signal of BLI-1 in the mid-L4, please show this data.

As the mid-L4 diffuse BLI-1::mNG signal is extremely faint by confocal imaging we have added images of the brighter diffuse BLI-2::mNG pattern to Supplementary Figure 4B.

These diffuse signals may represent BLI protein transiting the secretory pathway of underlying epidermal seam cells.

Please also show the localization of the N-terminal construct, which is stated to look like the C-terminal construct.

We have clarified (line 210) that most images and analysis shown are of the N-terminal mNG knock-in. We have added images of the C-terminal knock-in (ju1791) to Supplementary Figure 2B.

Comment 3.2 > Lines 210-219, 220-221: Where are the supporting images for these statements:

“Rows of BLI-1::mNG puncta were observed over almost all the lateral and dorsoventral cuticle, with mNG brightness varying depending on the region. Puncta in the head and the dorsal tail cuticle (overlying epidermal cells hyp5 and hyp10) were brighter than in other regions. BLI-1::mNG puncta overlying the edges of body muscle quadrants were also brighter than those in non-muscle-adjacent epidermis, suggesting that within each epidermal cell, deposition of BLI-1 might be modulated by local mechanical forces or signals from muscle.

“...other cuticle regions containing little or no BLI-1::mNG include the anterior tip of the head (external to hyp4), the hermaphrodite vulva, and the excretory pore.”

We have added images of these cuticle regions to the revised Supplementary Figure 2, including for BLI-1,-2 and -6 mNG knockin strains. We have indicated an example (red dashed bracket in Figure 4A) showing the elevated brightness of dorsoventral (muscle quadrant) versus lateral BLI-1::mNG puncta.

Comment 3.3 > Lines 265-266: Where are the pictures showing BLI-1 in gaps of DPY-13 signal?

We have cited Figure 3H for these pictures, however to clarify the relative localization we have added a representative line scan of BLI-1::mSc and DPY-13::mNG to Supplementary Figure 2F.

(4) Fourth, it would be helpful to make the manuscript more accessible to readers not familiar with C. elegans. For example, the L4 cuticle, nascent adult cuticle, and young adult cuticles are described, but stages examined are not clearly labelled in figures. A schematic of cuticle development, and definition of key terms such as alae and cuticle furrows early in the manuscript will be very helpful.

We have revised the introduction to better introduce these terms and have added a link to the cuticle section of *WormAtlas* (line 73). We have added statements to relevant Figure legends that all confocal and 3D SIM images are of young adults (within 24 h of the L4/adult molt), except for images in Figure 4 and in the Videos. In Figure 4 legend we have added additional detail on the L4 substages (based on vulval morphology) during which the BLI-1/2/6 patterns develop.

Minor comments:

> Figure 3, Figure 5—please avoid using red, as many readers are red-green color blind. Please use a color-blind friendly pairing, such as cyan and magenta.

We have replaced red channels with magenta in all colocalization images.

> Figure 1C. Authors have C terminus on left and N terminus on right (?) of protein. This goes against standard convention of how proteins are drawn out—N terminus is on left and C terminus is on right.

All our cartoons of proteins have N-termini on the left and C-termini on the right.

Reviewer #2 (Remarks to the Author):

Adams et al. present a study on the patterns of collagen localization in the cuticle of C. elegans. Using a variety of approaches, they suggest the presence of nanoscale patterning. I was tasked to provide a technical review for this manuscript and will focus my comments on the author's use and analysis of microscopy approaches.

This work utilized confocal, widefield, and 3DSIM microscopy combined with several fluorescence labeling strategies. In general, standard technical information and validations for complex microscopy experiments are lacking in this work. For example, the author's "nanoscale patterning" claim relies on the colocalization of fluorescence labels, but microscopy information, colocalization quantification, and controls are missing.

Comment 1 > *For standard microscopy information, the lateral and axial sampling is missing for all methods, the confocal microscope and widefield microscope's optical configuration (e.g., the numerical aperture), the exposure times are missing, and any background or flatfield corrections applied in the acquisition software are not reported. Without this information, it is not possible to evaluate the colocalization claims.*

We would like to clarify that the only widefield data presented are the low-resolution whole animal images of Pnlp-29-GFP in Figure 1B, and a new widefield/deconvolution image of DPY-3::mNG in Figure 5E. All other imaging used LSM800 or 900 laser scanning confocal or OMX 3D-SIM. We have added further imaging details to Methods, including measurements of the LSM PSFs using 100 nm diameter multicolor beads (Zeiss). LSM800 imaging used Planapo 100x/NA 1.40 OIL DIC M27 objective, field of view 63.89 x 63.89 μm (pixel xy dimensions 0.0425 μm), with PSF FWHM of ~330 nm in xy/1120 nm in z (488 nm laser) and 240 nm (650 nm laser). Images in the new Supplementary Figure 2 used LSM900 Airyscan SR imaging with 100x Planapo objective, FWHM 260 nm in xy and 560 nm. z stack intervals were 0.29 μm such that the entire cuticle can be imaged in 6 z slices (6 x 0.29 = 1.74 μm). All imaging used Zeiss Immersol 518F and #1.5 cover glass. Pinhole sizes were 0.45-0.72 Airy units. Laser power was 5-10 %. Detector gain was set at 500-700 depending on the brightness of the sample. Full details are in metadata for each image which will be deposited in a suitable repository upon publication, per journal instructions. No background or flatfield correction was applied.

For colocalization analyses, two valuable guides for evaluating colocalization are doi: 10.1152/ajpcell.00462.2010 and doi: 10.1038/s41596-020-0313-9. Quantifying the colocalization, through multiple approaches, using the 3D stacks would help strengthen the case, which can be performed using Fiji.

We thank the reviewer for the helpful advice and have used these metrics to compare degree of colocalization of the different protein markers, using Zeiss colocalization workspace tools. We have added charts of these standard quantitative colocalization metrics to new Supplementary Figure 2. We have added notes to the Discussion that BLI-1,2 and 6 display colocalization within the limits of the optical resolutions used.

Comment 2 > For the 3DSIM data, the authors do not present a multicolor calibration of the instrument using the specific refractive index matching oils and sample refractive index used in the experiments. The authors should describe the exact alignment and quality check protocols to ensure colocalization analysis is possible using the data. The modulation contrast to noise ratio (MCNR) and the estimated resolutions in XYZ are two useful parameters to report. Resolutions should be reported using diffraction-limited emitters during calibration and estimated from the data using Fourier ring correlation or Image decorrelation analysis. Both are available as Fiji plugins.

In addition to single-color bead samples for calibration, 3DSIM data for multicolor beads should be present to ensure that the recovered super-resolution images are of sufficient resolution and quality to support the author's colocalization claims.

We understand the reviewer's concerns regarding alignment of the OMX due to its multi-camera setup. The OMX SR super-resolution microscope employed in the study was a new install (January 2021). The manufacturer's calibration protocol is based on a proprietary GE calibration target slide, Image Alignment Calibration in softWoRx, and verification of alignment using diffraction-limited, multi-color, 100-nm TetraSpeck beads (Thermo Fisher). After calibration, the alignment files were used to correct all multi-color images in softWoRx using the 'OMXAlign' task. For 3D-SIM in the 488-channel using TetraSpeck beads, FWHM in XY varied from 90-110 nm, and 310 nm in Z, employing the same oil (1.518), glass (#1.5), and aqueous buffer (PBS) approximating the refractive index of buffers during experiments. The average MCNR reported by the 'SIMcheck' plugin (ImageJ) for 3D-SIM reconstructions presented in Figure 5 were typically > 8 in the 488-channel, and > 4 in the 568-channel, which according to plugin documentation is considered 'good to excellent', and 'low to moderate', respectively. The MCNR values for specific images are now reported in the Figure 5 legend.

3D-SIM data presented in this study was acquired throughout 2021, during which the system was tested and calibrated on three separate occasions by Leica (formerly Cytiva) experts during install, training, and annual service sessions. We frequently employed TetraSpeck bead slides to test the system for its alignment and found it to remain very stable.

These checks are essential, as the patterns in Figure 5D resemble "hammerstroke" noise, commonly present in noise-limited SIM reconstructions.

We agree that SIM-based reconstructions may introduce artefacts if images are noise-limited or over-processed. We have therefore added an additional image of furrow collagen (DPY-3::mNG) localization to Figure 5E, acquired with OMX widefield microscopy and processed with deconvolution software that does not involve SIM type reconstruction. This and additional images not shown reveal similar heterogeneity of collagen distribution within the furrow. While our furrow collagen (DPY-7::sfGFP) reconstructions passed quality control in MCNR, furrow collagen knockins in general reconstruct less well than strut collagens due to their higher background of diffuse fluorescence within the cuticle.

In addition, the 3D renderings in Figure 5 are a bit puzzling. Can the authors please explain the lack of detail in the axial dimension for Figures 5B and 5E?

We believe this reflects both the lower resolution of imaging in the z axis, and that struts are relatively uniform along this axis within the medial layer. By comparison, the DPY-7 'columns' are less uniform in the z-axis.

Beyond the microscopic methods, I have two additional points I would ask the authors to clarify:

Comment 3 > Why are Figures 6 and 7 presented with inverted lookup tables, while the rest of the manuscript is present with standard lookup tables?

In Figure 6 B-G, as well as in Figure 1A-D and Figure 4A-C, Figure 5A we present image data with standard lookup table (LUT) as well as higher magnification insets with inverted LUT, as the inverted LUTs may allow better visualization of fine details. Inverted LUTs are widely used to increase perceptual contrast as readers' visual systems are adapted to the white background of the page. Where possible we have used both LUTs to allow readers to compare the original and inverted images. In Figure 6H-J and Figure 7 we only included inverted LUT images due to space limitations when showing images of multiple genotypes at adequate resolution.

Comment 4 > For the image analysis, did the authors explore how varying the threshold around their chosen threshold altered their puncta counting results? The value should be robust to perturbations for the proposed analysis to be valid.

We agree with the reviewer that automated particle analysis should be somewhat robust to thresholds. Our automated particle analysis (using Fiji) requires a brightness threshold to binarize the image data, then an area threshold is applied. Varying the brightness threshold by +/- 20% did not significantly affect the results. We set the area threshold at 0.01–2 μm^2 to capture almost all struts that we would score as 'ground truth' by visual inspection. Our manual counts rely on additional criteria beyond size and shape, e.g. that struts typically form three rows per annulus and have reproducible spacing. Setting the particle area threshold below 0.01 μm^2 resulted in many small particles being counted as puncta, reflected in the wide base of the distributions in Fig 7D, which become bimodal if no area threshold is used. Area thresholds between 0.01 and 0.02 μm^2 return similar results. Thresholds above 0.02 μm^2 exclude puncta that would have been classified manually as small struts. In sum, automated particle counting is inherently somewhat sensitive to brightness and size thresholds and we have been careful to validate the sizes and numbers by manual counts.

Reviewer #1 (Remarks to the Author):

The revised manuscript has addressed all of my concerns.

I am particularly impressed with the rigor of the newly added quantification.

I am also pleased that the study has now placed the work in the context of other apical matrices and how the study advances our understanding of micro patterning these important, but understudied ECMS.

Reviewer #2 (Remarks to the Author):

I thank the authors for providing an extensive revision and answers to my questions.

Largely my concerns about the microscopy elements have been answered. I still suggest providing the calibration data (e.g. beads) to allow for readers to judge themselves if the precision required for co-localization at the nanoscale is met with the performed microscopy.

Reading through the revised manuscript, one thing that did stand out to be is the variety of statistical used without a rationale for each test.

I catalog what I could find here:

Figure 1: ordinary one-way ANOVA, Dunnett's post test.

Figure 2: both Kruskal-Wallis test and Dunn's post test and Ordinary one-way ANOVA and Šidák's post test are used.

Figure 6: Kruskal-Wallis test and Dunn's post test.

Figure 7: Mann-Whitney Test

Supplementary Figure 2: Kruskal-Wallis test and Dunn's post test

Supplementary Figure 3: ANOVA and Dunnett's post test

The only information provided on statistics is: "All statistical analysis used GraphPad Prism 10."

Can the author's please justify all statistical tests and update the methods?

REVIEWERS' COMMENTS

Reviewer #1 (Remarks to the Author):

The revised manuscript has addressed all of my concerns.

I am particularly impressed with the rigor of the newly added quantification.

I am also pleased that the study has now placed the work in the context of other apical matrices and how the study advances our understanding of micro patterning these important, but understudied ECMs.

We thank the reviewer for the supportive comments.

Reviewer #2 (Remarks to the Author):

I thank the authors for providing an extensive revision and answers to my questions.

Largely my concerns about the microscopy elements have been answered. I still suggest providing the calibration data (e.g. beads) to allow for readers to judge themselves if the precision required for co-localization at the nanoscale is met with the performed microscopy.

We thank the reviewer for the suggestion. For reasons of space we have not included the raw calibration images, given that standard PSF metrics (FWHM at different wavelengths) are normally considered sufficient to assess resolution. We have included some images of the raw calibration data in the source data files on Figshare.

Reading through the revised manuscript, one thing that did stand out to be is the variety of statistical used without a rationale for each test.

I catalog what I could find here:

Figure 1: ordinary one-way ANOVA, Dunnett's post test.

Figure 2: both Kruskal-Wallis test and Dunn's post test and Ordinary one-way ANOVA and Šidák's post test are used.

Figure 6: Kruskal-Wallis test and Dunn's post test.

Figure 7: Mann-Whitney Test

Supplementary Figure 2: Kruskal-Wallis test and Dunn's post test

Supplementary Figure 3: ANOVA and Dunnett's post test

The only information provided on statistics is: "All statistical analysis used GraphPad Prism 10."

Can the author's please justify all statistical tests and update the methods?

We have expanded the Methods section on statistics to justify the various tests, as follows:

Statistical Analysis and Reproducibility

Data sets were first tested for normality using the Kolmogorov-Smirnov test. Data passing this test were analyzed by parametric tests such as ordinary one-way ANOVA, with a post test determined by the experimental design (e.g. Šidák's multiple comparison test when comparing each experimental group to a control group, Dunn's multiple comparison test when comparing selected groups). Data sets not passing the normality test were analyzed using nonparametric tests i.e. Mann-Whitney test for comparing two data sets or Kruskal-Wallis test for > 2 comparisons. All statistical analysis used GraphPad Prism 10; exact P values are stated when reported by the software. All imaging and genetic analyses used stable genetically defined *C. elegans* strains that will be made available for reanalysis upon reasonable request. In general, at least three biological replicates per strain were analyzed independently (i.e. at least 3 animals per genotype imaged). Standard confocal fluorescence images are representative of 5-10 images per condition acquired over at least 3 imaging sessions. Electron microscopy sample sizes are noted below. 3D-SIM data are representative of at least 10 SIM data sets per condition acquired over 5-6 imaging sessions. Dot plots show the mean (red line), and SEM (orange error bar).